# GenMol: A Drug Discovery Generalist with Discrete Diffusion

Seul Lee [1]   Karsten Kreis [2]   Srimukh Prasad Veccham [2]   Meng Liu [2]
Danny Reidenbach [2]   Yuxing Peng [2]   Saee Paliwal [2]   Weili Nie* [2]   Arash Vahdat* [2]

## Abstract

Drug discovery is a complex process that involves multiple stages and tasks. However, existing molecular generative models can only tackle some of these tasks. We present *Generalist Molecular generative model* (GenMol), a versatile framework that uses only a *single* discrete diffusion model to handle diverse drug discovery scenarios. GenMol generates Sequential Attachment-based Fragment Embedding (SAFE) sequences through non-autoregressive bidirectional parallel decoding, thereby allowing the utilization of a molecular context that does not rely on the specific token ordering while having better sampling efficiency. GenMol uses fragments as basic building blocks for molecules and introduces *fragment remasking*, a strategy that optimizes molecules by regenerating masked fragments, enabling effective exploration of chemical space. We further propose *molecular context guidance* (MCG), a guidance method tailored for masked discrete diffusion of GenMol. GenMol significantly outperforms the previous GPT-based model in *de novo* generation and fragment-constrained generation, and achieves state-of-the-art performance in goal-directed hit generation and lead optimization. These results demonstrate that GenMol can tackle a wide range of drug discovery tasks, providing a unified and versatile approach for molecular design.

## 1. Introduction

Discovering molecules with the desired chemical profile is the core objective of drug discovery (Hughes et al., 2011). To achieve the ultimate goal of overcoming disease, a variety of drug discovery approaches have been established.

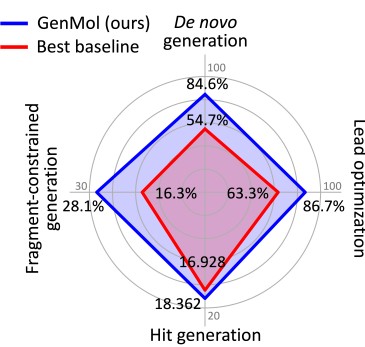

Figure 1: **Results on drug discovery tasks.** The values are quality, average quality, sum AUC top-10, and success rate for *de novo* generation, fragment-constrained generation, hit generation, and lead optimization, respectively. The "best baseline" refers to multiple best-performing task-specific models among prior works.

For example, fragment-constrained molecule generation is a popular strategy for designing new drug candidates under the constraint of preserving a certain molecular substructure already known to exhibit a particular bioactivity (Murray & Rees, 2009). Furthermore, real-world drug discovery pipelines are not a single stage but consist of several key stages, such as hit generation and lead optimization (Hughes et al., 2011). A drug discovery process that leads to the finding of drug candidates that can enter clinical trials should consider all of these different scenarios.

Generative models have emerged as a promising methodology to accelerate labor-intensive drug discovery pipelines (Olivecrona et al., 2017; Jin et al., 2018; Yang et al., 2021; Lee et al., 2023), but previous molecular generative models have a common limitation: they focus on only one or two of the drug discovery scenarios. They either cannot be applied to multiple tasks or require expensive modifications including retraining of a specific architecture for each task (Yang et al., 2020; Guo et al., 2023). Recently, SAFE-GPT (Noutahi et al., 2024) has been proposed to address this problem by formulating several molecular tasks as a fragment-constrained generation task, solved by sequence completion. SAFE-GPT uses Sequential Attachment-based Fragment Embedding (SAFE) of molecules, which represents a molecule as an unordered sequence of Simplified Molecular Input Line Entry System (SMILES) (Weininger, 1988) fragment blocks. However,

---

[1]KAIST. Work during an internship at NVIDIA. [2]NVIDIA. *Equal advising. Correspondence to: Seul Lee <seul.lee@kaist.ac.kr>.

*Proceedings of the 42nd International Conference on Machine Learning*, Vancouver, Canada. PMLR 267, 2025. Copyright 2025 by the author(s).

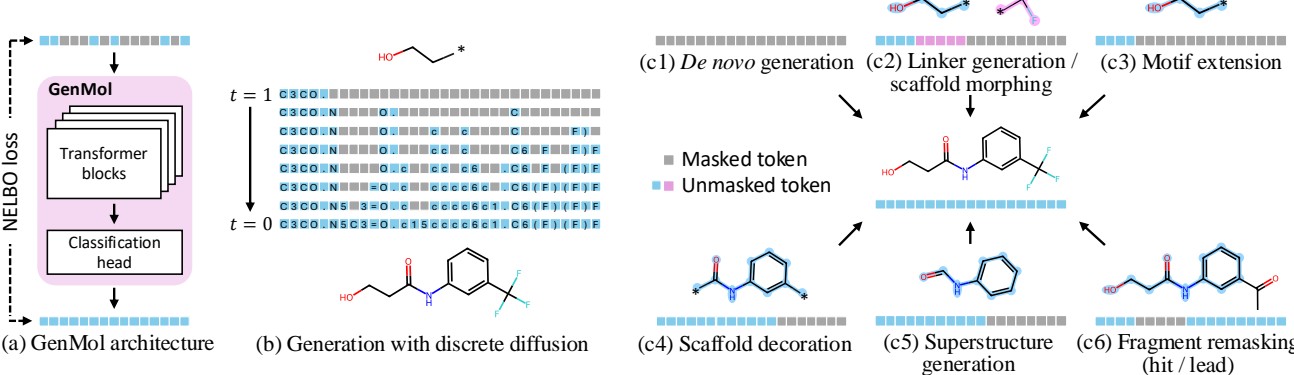

Figure 2: **(a) GenMol architecture.** GenMol adopts the BERT architecture and is trained with the NELBO loss of masked discrete diffusion. **(b) Generation process of GenMol.** Under masked discrete diffusion, GenMol completes a molecule by simulating backward in time and predicting masked tokens at each time step $t$ until all tokens are unmasked. **(c) Illustration of various drug discovery tasks that can be performed by GenMol.** GenMol is endowed with the ability to easily perform (c1) *de novo* generation, (c2-c5) fragment-constrained generation, and (c6) fragment remasking that can be applied to goal-directed hit generation and lead optimization.

since all sequence-based molecular representations including SMILES and SAFE assume an ordering of tokens based on heuristic rules such as depth-first search (DFS), autoregressive models that operate in a left-to-right order like GPT are unnatural for processing and generating molecular sequences. In addition, the autoregressive decoding scheme limits the computational efficiency of the model and makes it challenging to introduce guidance during generation. Moreover, SAFE-GPT relies on finetuning under expensive reinforcement learning (RL) objectives to be applied to goal-directed molecule generation.

To tackle these limitations, we propose *Generalist Molecular generative model* (GenMol), a versatile molecular generation framework that is endowed with the ability to handle diverse scenarios that can be encountered in the multifaceted drug discovery pipeline (Figure 2(c)). GenMol adopts a masked discrete diffusion framework (Austin et al., 2021; Sahoo et al., 2024; Shi et al., 2024) with the BERT architecture (Devlin et al., 2019) to generate SAFE molecular sequences, thereby enjoying several advantages: (i) Discrete diffusion allows GenMol to exploit a molecular context that does not rely on the specific ordering of tokens and fragments by bidirectional attention. (ii) The non-autoregressive parallel decoding improves GenMol's computational efficiency (Figure 2(b)). (iii) Discrete diffusion enables GenMol to explore chemical space with a simple yet effective remasking strategy. We propose *fragment remasking* (Figure 2(c6) and Figure 3), a strategy to optimize molecules by replacing certain fragments in a given molecule with masked tokens, from which diffusion generates new fragments. Utilizing fragments as the explorative unit instead of individual tokens is more in line with chemists' intuition during optimization in drug discovery, and GenMol can effectively and efficiently explore the vast chemical space to find chemical optima. (iv) Discrete diffu-

sion also makes it possible to apply guidance during generation based on the entire sequence. To this end, we propose *molecular context guidance* (MCG), a guidance method to improve the performance of GenMol by calibrating its predictions with information in a given molecular context.

We experimentally validate GenMol on a wide range of molecule generation tasks that simulate real-world drug discovery problems, including *de novo* generation, fragment-constrained generation, goal-directed hit generation, and goal-directed lead optimization. Across extensive experiments, GenMol outperforms existing methods by a large margin (Figure 1). Note that the best baseline results shown in Figure 1 are not the results of a single model, but of multiple task-specific models. These results demonstrate GenMol's potential as a versatile tool that can be used throughout the drug discovery pipeline.

We summarize our contributions as follows:

- We introduce GenMol, a framework for unified and versatile molecule generation by building masked discrete diffusion that generates SAFE molecular sequences.
- We propose fragment remasking, an effective strategy for exploring chemical space using molecular fragments as the unit of exploration.
- We propose MCG, a guidance scheme for GenMol to effectively utilize molecular context information.
- We validate the efficacy and versatility of GenMol on a wide range of drug discovery tasks.

## 2. Related Work

**Discrete diffusion.** There has been steady progress in applying discrete diffusion for discrete data generation, especially in NLP tasks (Hoogeboom et al., 2021; Austin et al., 2021; He et al., 2023; Zheng et al., 2024; Lou et al.,

2024; Sahoo et al., 2024). This is mainly due to their non-autoregressive generation property, which leads to a potential for better modeling long-rang bidirectional dependencies and acclerating sampling speed, and their flexible design choices in training, sampling, and controllable generation (Sahoo et al., 2024). Notably, D3PM (Austin et al., 2021) introduced a general framework with a Markov forward process represented by transition matrices, and a transition matrix with an absorbing state corresponds to the masked language modeling (MLM) such as BERT (Devlin et al., 2019). Campbell et al. (2022) proposed a continuous-time framework for discrete diffusion models based on the continuous-time Markov chain (CTMC) theory. SEDD (Lou et al., 2024) introduced a denoising score entropy loss that extends score matching to discrete diffusion models. Sahoo et al. (2024) and Shi et al. (2024) proposed simple masked discrete diffusion frameworks, with the training objective being a weighted average of MLM losses across different diffusion time steps.

Recently, a few works have applied discrete diffusion for molecular generation. For instance, DiGress (Vignac et al., 2023) followed the D3PM framework to generate molecular graphs with categorical node and edge attributes. Other works (Zhang et al., 2023; Lin et al., 2024; Hua et al., 2024) focused on the 3D molecular structure generation, where they used discrete diffusion for atom type generation and continuous diffusion for atom position generation. However, none of them applied discrete diffusion for molecular sequence generation that can serve as a generalist foundation model for solving various downstream tasks.

**Fragment-based drug discovery.** Fragment-based molecular generative models refer to a class of methods that reassemble existing molecular substructures (i.e., fragments) to generate new molecules. They have been consider as an effective drug discovery approach as (i) assembling fragments simplifies the generation process and improves chemical validity and (ii) the unit that determines biochemical effect of a molecule is a fragment rather than an individual atom (Li, 2020). A line of works (Jin et al., 2020; Maziarz et al., 2021; Kong et al., 2022; Geng et al., 2023) used graph-based VAEs to generate novel molecules conditioned on discovered substructures. Xie et al. (2020) proposed to progressively add or delete fragments of molecular graphs using Markov chain Monte Carlo (MCMC) sampling. Yang et al. (2021) and Powers et al. (2023) used a reinforcement learning (RL) framework and classification, respectively, to progressively add fragments to the incomplete molecule. Graph-based genetic algorithms (GAs) (Jensen, 2019; Tripp & Hernández-Lobato, 2023) is a strong approach that decomposes parent molecules into fragments that are combined to generate an offspring molecule. However, since their generation is from random combinations of existing fragments with

a local mutation of a small probability, they suffer from limited exploration in the chemical space. More recently, $f$-RAG (Lee et al., 2024a) introduced a fragment-level retrieval framework that augments the pre-trained molecular language model SAFE-GPT (Noutahi et al., 2024), where retrieving fragments from dynamically updated fragment vocabulary largely improves the exploration-exploitation trade-off. However, $f$-RAG still needs to train an information fusion module before adapting to various goal-oriented generation tasks.

## 3. Background

### 3.1. Masked Diffusion

Masked diffusion models (Sahoo et al., 2024; Shi et al., 2024) are a simple and effective class of discrete diffusion models, and we follow MDLM (Sahoo et al., 2024) to define our masked diffusion. Formally, we define $\boldsymbol{x}$ as a sequence of $L$ tokens, each of which, denoted as $\boldsymbol{x}^l$, is a one-hot vector with $K$ categories (i.e., $\boldsymbol{x}_i^l \in \{0,1\}^K$ and $\sum_{i=1}^K \boldsymbol{x}_i^l = 1$). Without loss of generality, we assume the $K$-th category represents the masking token, whose one-hot vector is denoted by $\mathbf{m}$ (i.e., $\mathbf{m}_K = 1$). We also define $\text{Cat}(\cdot; \boldsymbol{\pi})$ as a categorical distribution with a probability $\boldsymbol{\pi} \in \Delta^K$, where $\Delta^K$ represents the simplex over $K$ categories.

The forward masking process independently interpolates the probability mass between each token in clean data sequence $\boldsymbol{x}^l$ and the masking token $\mathbf{m}$, defined as

$$q(\boldsymbol{z}_t^l | \boldsymbol{x}^l) = \text{Cat}(\boldsymbol{z}_t^l; \alpha_t \boldsymbol{x}^l + (1 - \alpha_t)\mathbf{m}), \quad (1)$$

where $\boldsymbol{z}_t^l$ denotes the $l$-th token in the noisy data sequence at the time step $t \in [0,1]$, and $\alpha_t \in [0,1]$ denotes the masking ratio that is monotonically decreasing function of $t$, with $\alpha_0 = 1$ to $\alpha_1 = 0$. Accordingly, at time step $t = 1$, $\boldsymbol{z}_t$ becomes a sequence of all masked tokens.

The reverse unmasking process inverts the masking process and independently infers each token of the less masked data $\boldsymbol{z}_s$ from more masked data $\boldsymbol{z}_t$ with $s < t$, which is given by

$$p_\theta(\boldsymbol{z}_s^l | \boldsymbol{z}_t^l) = \begin{cases} \text{Cat}(\boldsymbol{z}_s^l; \boldsymbol{z}_t^l) & \boldsymbol{z}_t^l \neq \mathbf{m} \\ \text{Cat}(\boldsymbol{z}_s^l; \frac{(1-\alpha_s)\mathbf{m}+(\alpha_s-\alpha_t)\boldsymbol{x}_\theta^l(\boldsymbol{z}_t,t)}{1-\alpha_t}) & \boldsymbol{z}_t^l = \mathbf{m}, \end{cases} \quad (2)$$

where $\boldsymbol{x}_\theta(\boldsymbol{z}_t, t)$ is a denoising network that takes the noisy data sequence $\boldsymbol{z}_t$ as input and predicts $L$ probability vectors for the clean data sequence. This parameterization designs the reverse process such that it does not change unmasked tokens. To train the denoising network $\boldsymbol{x}_\theta(\boldsymbol{z}_t, t)$, the training objective, which implicitly approximates the negative ELBO (Sohl-Dickstein et al., 2015), is given by

$$\mathcal{L}_{\text{NELBO}} = \mathbb{E}_q \int_0^1 \frac{\alpha_t'}{1 - \alpha_t} \sum_l \log\langle \boldsymbol{x}_\theta^l(\boldsymbol{z}_t, t), \boldsymbol{x}^l \rangle \mathrm{d}t, \quad (3)$$

which is the weighted average of MLM losses (i.e., cross-entropy losses) over time steps.

## 3.2. SAFE Molecular Representation

Simplified Molecular Input Line Entry System (SMILES) (Weininger, 1988) is the most widely used molecular string representation, but it relies on a heuristic depth-first search (DFS) that traverses the atoms of a molecule. Therefore, atoms that are close in molecular structure can be tokens that are very far apart in molecular sequence, and thus it is not straightforward to perform fragment-constrained molecular generation with SMILES.

Sequential Attachment-based Fragment Embedding (SAFE) (Noutahi et al., 2024) has been proposed to alleviate this problem. SAFE represents molecules as an unordered sequence of fragment blocks, thereby casting molecular design into sequence completion. SAFE is a non-canonical SMILES in which the arrangement of SMILES tokens corresponding to the same molecular fragment is consecutive. Molecules are decomposed into fragments by the BRICS algorithm (Degen et al., 2008) and the fragments are concatenated using a dot token (".") while preserving their attachment points. SAFE is permutation-invariant on fragments, i.e., the order of fragments within a SAFE string does not change the molecular identity.

## 4. Method

We introduce GenMol, a universal molecule generation framework that can solve various drug discovery tasks. We first introduce the construction of the discrete diffusion framework on the SAFE representation in Section 4.1. Next, we describe the goal-oriented exploration strategy of GenMol with fragment remasking in Section 4.2. Finally, we describe MCG, a guidance scheme of GenMol by partially masking the given molecular context, in Section 4.3.

### 4.1. Masked Diffusion for Molecule Generation

We adopt masked discrete diffusion to generate SAFE sequences and establish a flexible and efficient molecule generation framework. Concretely, GenMol uses the BERT architecture (Devlin et al., 2019) as the denoising network $\boldsymbol{x}_\theta$ and the training scheme of MDLM. Utilizing discrete diffusion instead of an autoregressive model is more in line with the SAFE representation and has several advantages. First, due to the bidirectional attention in BERT, GenMol can leverage parallel decoding where all tokens are decoded simultaneously under discrete diffusion (Figure 5). As SAFE is fragment order-insensitive, this allows GenMol to predict masked tokens without relying on a specific ordering of generation while considering the entire molecule. The non-autoregressive parallel decoding scheme also improves

GenMol's efficiency. Furthermore, the discrete diffusion framework enables GenMol to explore the neighborhood of a given molecule with a remasking strategy.

At each masked index $l$, GenMol samples $\boldsymbol{z}_s^l$ based on the reverse unmasking process $p_\theta^l := p_\theta(\boldsymbol{z}_s^l | \boldsymbol{z}_t)$ specified by:

$$p_{\theta,i}^l = \frac{\exp\left(\log \boldsymbol{x}_{\theta,i}^l(\boldsymbol{z}_t, t)/\tau\right)}{\sum_{j=1}^K \exp\left(\log \boldsymbol{x}_{\theta,j}^l(\boldsymbol{z}_t, t)/\tau\right)} \text{ for } i = 1, \cdots, K, (4)$$

where $\log \boldsymbol{x}_{\theta,i}^l(\boldsymbol{z}_t, t)$ is the logit predicted by the model and $\tau$ is the softmax temperature. All masked tokens are predicted in a parallel manner and GenMol confirms the top-$N$ confident predictions with additional randomness $r$ following Chang et al. (2022), where $N$ is the number of tokens to unmask at each time step. Trade-offs between molecular quality and diversity often arise in drug discovery, and GenMol can balance them through the softmax temperature $\tau$ and the randomness $r$. Further details about confidence-based sampling is provided in Section B.

### 4.2. Exploration in Chemical Space with GenMol

To perform goal-directed molecular optimization tasks, we propose a simple yet effective generation method (Figure 3) that consists of three steps: (1) fragment scoring, (2) fragment attaching, and (3) fragment remasking.

**Fragment scoring.** We start with constructing a fragment vocabulary. A set of $D$ molecules $\mathcal{D} = \{(\boldsymbol{x}_d, y(\boldsymbol{x}_d))\}_{d=1}^D$, where $y(\boldsymbol{x}_d)$ is the target property of molecule $\boldsymbol{x}_j$, is decomposed into a set of $F$ fragments $\mathcal{F} = \{\boldsymbol{f}_k\}_{k=1}^F$ using a predefined decomposition rule. We define the score of fragment $\boldsymbol{f}_k$ following Lee et al. (2024a) as:

$$y(\boldsymbol{f}_k) = \frac{1}{|\mathcal{S}(\boldsymbol{f}_k)|} \sum_{\boldsymbol{x} \in \mathcal{S}(\boldsymbol{f}_k)} y(\boldsymbol{x}), \quad (5)$$

where $\mathcal{S}(\boldsymbol{f}_k) = \{\boldsymbol{x} : \boldsymbol{f}_k \text{ is a subgraph of } \boldsymbol{x}\}$ and the top-$V$ fragments based on Eq. (5) are selected as the vocabulary.

**Fragment attaching.** During generation, a molecule $\boldsymbol{x}_{\text{init}}$ is first generated by randomly selecting two fragments from the vocabulary and attaching them. Fragments or functional groups influence the chemical properties of a molecule and therefore, fragments that commonly occur in molecules with desirable properties are likely to carry them to new molecules. However, with fragment attaching alone, the model cannot generate new fragments that are not included in the initial vocabulary, resulting in suboptimal exploration in chemical space.

**Fragment remasking.** Therefore, utilizing discrete diffusion of GenMol, we propose *fragment remasking*, an

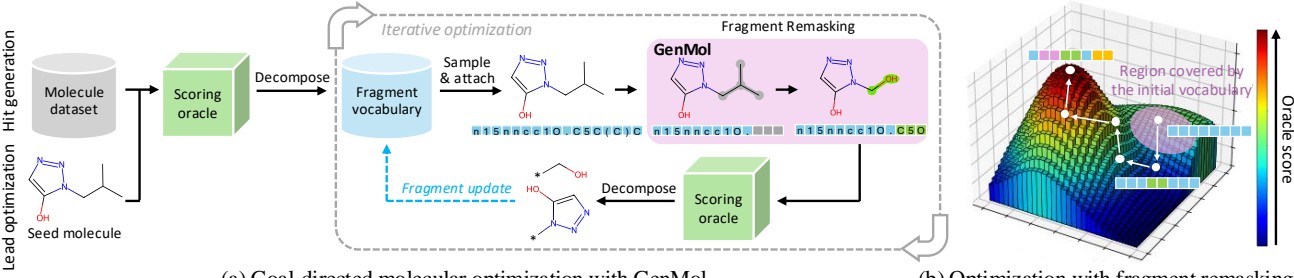

(a) Goal-directed molecular optimization with GenMol      (b) Optimization with fragment remasking

Figure 3: **(a) Goal-directed hit generation and lead optimization process with GenMol.** An initial fragment vocabulary is constructed by decomposing an existing molecular dataset (hit generation) or a seed molecule (lead optimization). Two fragments are randomly sampled from the vocabulary and attached, and GenMol performs fragment remasking. The fragment vocabulary is updated with the generated molecules for the next iteration. **(b) Illustration of the molecular optimization trajectory with fragment remasking.** With fragment remasking, GenMol can explore beyond the initial fragment vocabulary to find chemical optima.

effective strategy to explore the neighborhood of a given molecule in chemical space to find optimal molecules. Discrete diffusion allows GenMol to mask and re-predict some tokens of a given molecule, making neighborhood exploration simple and straightforward. However, although there exist some works that apply token-wise remasking to protein sequence optimization (Hayes et al., 2024; Gruver et al., 2024), it would be suboptimal to naively adopt the same strategy for small molecule optimization. This is because each token in a SAFE (or SMILES) sequence represents a single atom or bond, and masking them individually results in localized and ineffective exploration. The units that carry information about molecular properties are fragments or functional groups, not individual atoms or bonds (Li, 2020), and thus fragment remasking modifies a molecule by randomly re-predicting a fragment.

Given $\boldsymbol{x}_{\text{init}}$, $\boldsymbol{x}_{\text{mask}}$ is constructed by randomly selecting one of the fragments of $\boldsymbol{x}_{\text{init}}$ and replacing it with a mask chunk. Here, a finer decomposition rule than the one that constructed the vocabulary is used, allowing fragment remasking to operate at a fine-grained level. GenMol then generates $\boldsymbol{x}_{\text{new}}$ by iteratively unmasking $\boldsymbol{x}_{\text{mask}}$. The proposed fragment remasking can also be viewed as a mutation operation in GA where the mutation is performed at the fragment-level rather than at the atom- or bond-level. $\boldsymbol{x}_{\text{new}}$ is decomposed and scored by Eq. (5), and the fragment vocabulary is dynamically updated by selecting the top-$V$ fragments, allowing exploration beyond the initial fragments. We summarize the goal-directed generation process of GenMol with fragment remasking in Algorithm 1.

Instead of using a fixed length, the lengths of the fragment mask chunks are sampled from a predefined distribution, e.g., the distribution of fragment lengths in the training set. The length of the fragment depends on the decomposition rule used, and this strategy allows GenMol to automatically adjust the length based on the rule. This also ensures that GenMol generates fragments of varying lengths, offering users better controllability over molecule generation.

---

**Algorithm 1** Goal-directed Molecular Optimization of GenMol

**Input:** A set of molecules $\mathcal{D}$, vocabulary size $V$, decomposition rule for fragment vocabulary $R_{\text{vocab}}$, decomposition rule for fragment remasking $R_{\text{remask}}$, number of generations $G$

Set $\mathcal{F} \leftarrow$ fragments obtained by decomposing $\mathcal{D}$ with $R_{\text{vocab}}$
Set $\mathcal{V} \leftarrow$ top-$V$ fragments of $\mathcal{F}$ (Eq. 5)
Set $p_{\text{len}} \leftarrow$ fragment length distribution of $\mathcal{D}$ based on $R_{\text{remask}}$
Set $\mathcal{M} \leftarrow \emptyset$
**while** $|\mathcal{M}| < G$ **do**
    Select and attach two fragments from $\mathcal{V}$ to get $\boldsymbol{x}_{\text{init}}$
    Sample the fragment length $m \sim p_{\text{len}}$
    Select one of the fragments of $\boldsymbol{x}_{\text{init}}$ based on $R_{\text{remask}}$ and replace it with $m$ mask tokens to get $\boldsymbol{x}_{\text{mask}}$
    Generate $\boldsymbol{x}_{\text{new}}$ by iteratively unmasking $\boldsymbol{x}_{\text{mask}}$ with GenMol
    Update $\mathcal{M} \leftarrow \mathcal{M} \cup \{\boldsymbol{x}_{\text{new}}\}$
    Decompose $\boldsymbol{x}_{\text{new}}$ into fragments $\{\boldsymbol{f}_1, \boldsymbol{f}_2, \dots\}$ with $\mathcal{R}_{\text{vocab}}$
    Update $\mathcal{V} \leftarrow$ top-$V$ fragments from $\mathcal{V} \cup \{\boldsymbol{f}_1, \boldsymbol{f}_2, \dots\}$
**end while**
**Output:** Generated molecules $\mathcal{M}$

---

Fragment remasking can also be interpreted as Gibbs sampling (Geman & Geman, 1984). Assuming a SAFE molecular sequence $\boldsymbol{x}$ is comprised of $F$ fragments, we can represent $\boldsymbol{x}$ as a set of the fragments $\{\boldsymbol{f}_k\}_{k=1}^{F}$, where $\boldsymbol{f}_k$ denotes an attachment point-assigned SAFE fragment. To sample a molecule $\boldsymbol{x}$ from $p(\boldsymbol{x}) = p(\boldsymbol{f}_1, \dots, \boldsymbol{f}_F)$, fragment remasking repeats the process of uniformly selecting the index $k$ and then sampling $\boldsymbol{f}_k$ from $p(\boldsymbol{f}_k | \boldsymbol{f}_{\backslash k})$, where $\boldsymbol{f}_{\backslash k}$ denotes $\boldsymbol{f}_1, \dots, \boldsymbol{f}_F$ but with $\boldsymbol{f}_k$ omitted. This is equivalent to performing Gibbs sampling with the Markov kernel $p(\boldsymbol{f}_k | \boldsymbol{f}_{\backslash k})$, allowing GenMol to perform a random walk in the neighborhood of the given molecule $\boldsymbol{x}$.

### 4.3. Molecular Context Guidance

Inspired by autoguidance (Karras et al., 2024), we propose *molecular context guidance* (MCG), a guidance method tailored for masked discrete diffusion of GenMol. Karras et al. (2024) generalized classifier-free guidance (CFG) (Ho & Salimans, 2021) and proposed to extrapolate

Table 1: *De novo* **molecule generation results.** The results are the means and the standard deviations of 3 runs. $N$, $\tau$, and $r$ is the number of tokens to unmask at each time step, the softmax temperature, and the randomness, respectively. The best results are highlighted in bold.

| Method | Validity (%) | Uniqueness (%) | Quality (%) | Diversity | Sampling time (s) |
|---|---|---|---|---|---|
| SAFE-GPT | $94.0 \pm 0.4$ | $\mathbf{100.0} \pm 0.0$ | $54.7 \pm 0.3$ | $0.879 \pm 0.001$ | $27.7 \pm 0.1$ |
| GenMol w/o conf. sampling | $96.7 \pm 0.3$ | $99.3 \pm 0.2$ | $53.8 \pm 1.7$ | $0.896 \pm 0.001$ | $25.4 \pm 1.6$ |
| GenMol ($\tau = 0.5, r = 0.5$) | | | | | |
| $N = 1$ | $\mathbf{100.0} \pm 0.0$ | $99.7 \pm 0.1$ | $\mathbf{84.6} \pm 0.8$ | $0.818 \pm 0.001$ | $21.1 \pm 0.4$ |
| $N = 2$ | $97.6 \pm 0.7$ | $99.5 \pm 0.2$ | $76.2 \pm 1.3$ | $0.843 \pm 0.002$ | $12.2 \pm 0.6$ |
| $N = 3$ | $95.6 \pm 0.5$ | $99.0 \pm 0.1$ | $67.1 \pm 0.7$ | $0.861 \pm 0.001$ | $\mathbf{10.1} \pm 0.2$ |
| GenMol ($N = 1$) | | | | | |
| $\tau = 0.5, r = 0.5$ | $\mathbf{100.0} \pm 0.0$ | $99.7 \pm 0.1$ | $\mathbf{84.6} \pm 0.8$ | $0.818 \pm 0.001$ | $21.1 \pm 0.4$ |
| $\tau = 0.5, r = 1.0$ | $99.7 \pm 0.1$ | $\mathbf{100.0} \pm 0.0$ | $83.8 \pm 0.5$ | $0.832 \pm 0.001$ | $20.5 \pm 0.6$ |
| $\tau = 1.0, r = 1.0$ | $99.8 \pm 0.1$ | $\mathbf{100.0} \pm 0.1$ | $79.1 \pm 0.9$ | $0.845 \pm 0.002$ | $21.9 \pm 0.6$ |
| $\tau = 1.0, r = 10.0$ | $99.8 \pm 0.1$ | $99.6 \pm 0.1$ | $63.0 \pm 0.4$ | $0.882 \pm 0.003$ | $21.5 \pm 0.5$ |
| $\tau = 1.5, r = 10.0$ | $95.6 \pm 0.3$ | $98.3 \pm 0.2$ | $39.7 \pm 0.5$ | $\mathbf{0.911} \pm 0.004$ | $20.9 \pm 0.5$ |

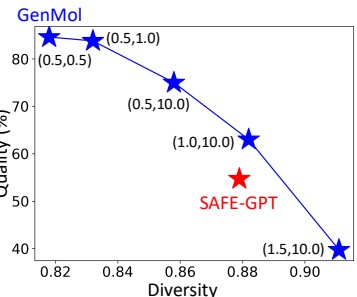

Figure 4: **The quality-diversity trade-off in *de novo* generation** with different values of $(\tau, r)$.

between the predictions of two denoising networks:

$$D^{(w)}(\boldsymbol{z}_s | \boldsymbol{z}_t, \boldsymbol{y}) = w D_1(\boldsymbol{z}_s | \boldsymbol{z}_t, \boldsymbol{y}) + (1-w) D_0(\boldsymbol{z}_s | \boldsymbol{z}_t, \boldsymbol{y}), \quad (6)$$

where $D_1$ and $D_0$ are a high-quality model and a poor model, respectively, $\boldsymbol{y}$ is the condition to perform the guidance on, and $w > 1$ is the guidance scale. Typically the same network $D_\theta$ is used for $D_1$ and $D_0$ with additional degradations applied to $D_0$, such as corrupted input, e.g., CFG sets $D_1 = D_\theta(\boldsymbol{z}_s | \boldsymbol{z}_t, \boldsymbol{y})$ and $D_0 = D_\theta(\boldsymbol{z}_s | \boldsymbol{z}_t, \emptyset)$. The idea of autoguidance is that the weaker version of the same model amplifies the errors, and thus emphasizing the output of $D_1$ over $D_0$ by setting $w > 1$ eliminates these errors.

However, the above guidance method has only been applied to continuous diffusion and its application to discrete diffusion has remained unexplored. On the other hand, Nisonoff et al. (2024) proposed a CFG scheme for continuous-time Markov chains (CTMCs) (Campbell et al., 2022), a subclass of discrete diffusion models based on a continuous-time formulation. As GenMol is trained under the MDLM framework which can be interpreted as a CTMC (Sahoo et al., 2024), we introduce autoguidance for the MDLM formulation. Specifically, MCG sets $D_1$ as the original prediction of the denoiser $\boldsymbol{x}_\theta$ and $D_0$ as the prediction with partially corrupted input, replacing the logits in Eq. (4) by

$$\log \boldsymbol{x}_{\theta,i}^{(w),l}(\boldsymbol{z}_t, t) := w \log \boldsymbol{x}_{\theta,i}^l(\boldsymbol{z}_t, t) + (1-w) \log \boldsymbol{x}_{\theta,i}^l(\tilde{\boldsymbol{z}}_t, t), \quad (7)$$

where $\tilde{\boldsymbol{z}}_t$ is constructed by masking $\gamma \cdot 100\%$ of the tokens in $\boldsymbol{z}_t$. We provide the derivation in Section C. Intuitively, Eq. (7) compares two outputs from a single GenMol model, with good and poor input, respectively. Specifically, the good input is a given partially masked sequence, which is further masked by $\gamma \cdot 100\%$ to yield poor input, and the two resulting logits are compared to calibrate GenMol's predictions. Using MCG, GenMol can fully utilize the given molecular context information in fragment-constrained and goal-directed generation.

## 5. Experiments

GenMol is trained on the SAFE dataset (Noutahi et al., 2024), which combines molecules from ZINC (Irwin et al., 2012) and UniChem (Chambers et al., 2013). We emphasize that a single GenMol checkpoint is used to perform the following tasks without any additional finetuning specific to each task. We first conduct experiments on *de novo* molecule generation in Section 5.1. Next, we conduct experiments on fragment-constrained molecule generation tasks in Section 5.2. We then examine GenMol's ability to perform goal-directed hit generation and goal-directed lead optimization in Section 5.3 and Section 5.4, respectively. We perform ablation studies in Section 5.5.

### 5.1. *De Novo* Generation

**Setup.** In *de novo* generation, the goal is to generate valid, unique, and diverse molecules. We generate 1,000 molecules and evaluate them with the following metrics, following Noutahi et al. (2024). **Validity** is the fraction of generated molecules that are chemically valid. **Uniqueness** is the fraction of valid molecules that are unique. **Diversity** is defined as the average pairwise Tanimoto distance between the Morgan fingerprints of the generated molecules. We further introduce **quality**, the fraction of valid, unique, drug-like, and synthesizable molecules, to provide a single metric that evaluates the ability to generate chemically reasonable and unique molecules. Here, *drug-like* and *synthesizable* molecules are defined as those satisfying quantitative estimate of drug-likeness (QED) (Bickerton et al., 2012) $\geq 0.6$ and synthetic accessibility (SA) (Ertl & Schuffenhauer, 2009) $\leq 4$, respectively, following Jin et al. (2020). Further details are provided in Section D.3.

**Results.** The results are shown in Table 1. GenMol w/o conf. sampling is a GenMol that uses the standard diffusion sampling (Austin et al., 2021; Sahoo et al., 2024) instead of confidence sampling. SAFE-GPT (Noutahi et al., 2024), GenMol w/o conf. sampling, and GenMol all show a near-perfect uniqueness, while GenMol significantly out-

Table 2: **Fragment-constrained molecule generation results.** The results are the means and the standard deviations of 3 runs. The best results are highlighted in bold.

| Method | Task | Validity (%) | Uniqueness (%) | Quality (%) | Diversity | Distance |
|---|---|---|---|---|---|---|
| SAFE-GPT | Linker design | 76.6 ± 5.1 | 82.5 ± 1.9 | 21.7 ± 1.1 | 0.545 ± 0.007 | 0.541 ± 0.006 |
| | Scaffold morphing | 58.9 ± 6.8 | 70.4 ± 5.7 | 16.7 ± 2.3 | 0.514 ± 0.011 | 0.528 ± 0.009 |
| | Motif extension | **96.1** ± 1.9 | 66.8 ± 1.2 | 18.6 ± 2.1 | 0.562 ± 0.003 | 0.665 ± 0.006 |
| | Scaffold decoration | **97.7** ± 0.3 | 74.7 ± 2.5 | 10.0 ± 1.4 | 0.575 ± 0.008 | 0.625 ± 0.009 |
| | Superstructure generation | 95.7 ± 2.0 | 83.0 ± 5.9 | 14.3 ± 3.7 | 0.573 ± 0.028 | **0.776** ± 0.036 |
| GenMol | Linker design | **100.0** ± 0.0 | **83.7** ± 0.5 | **21.9** ± 0.4 | **0.547** ± 0.002 | **0.563** ± 0.003 |
| | Scaffold morphing | **100.0** ± 0.0 | **83.7** ± 0.5 | **21.9** ± 0.4 | **0.547** ± 0.002 | **0.563** ± 0.003 |
| | Motif extension | 82.9 ± 0.1 | **77.5** ± 0.1 | **30.1** ± 0.4 | **0.617** ± 0.002 | **0.682** ± 0.001 |
| | Scaffold decoration | 96.6 ± 0.8 | **82.7** ± 1.8 | **31.8** ± 0.5 | **0.591** ± 0.001 | **0.651** ± 0.001 |
| | Superstructure generation | **97.5** ± 0.9 | **83.6** ± 1.0 | **34.8** ± 1.0 | **0.599** ± 0.009 | 0.762 ± 0.007 |

performs the other two in terms of validity, quality, and sampling time. Thanks to the non-autoregressive parallel decoding scheme, GenMol can predict multiple tokens simultaneously and shows much faster sampling as $N$, the number of tokens to unmask at each time step, increases. Notably, GenMol with $N = 3$ shows higher quality than SAFE-GPT and GenMol w/o conf. sampling with 2.5x shorter sampling time and comparable diversity. Furthermore, GenMol can balance between quality and diversity by adjusting the values of the softmax temperature $\tau$ and the randomness $r$ of the confidence sampling. This balance is also shown in Figure 4, demonstrating that GenMol generates molecules along the Pareto frontier of the quality-diversity trade-off. Further analysis on quality and diversity are provided in Section E.2, Section E.3 and Section E.4.

## 5.2. Fragment-constrained Generation

**Setup.** In fragment-constrained generation, the goal is to complete a molecule given a set of fragments, a frequently encountered setting in real-world drug discovery. We use the benchmark of Noutahi et al. (2024), which uses input fragments extracted from 10 known drugs to perform **linker design**, **scaffold morphing**, **motif extension**, **scaffold decoration**, and **superstructure generation**. In addition to **validity**, **uniqueness**, **diversity** and **quality** introduced in Section 5.1, **distance**, the average Tanimoto distance between the original and generated molecules, is also measured. 100 molecules are generated for each drug and averaged. Further details are provided in Section D.4.

**Results.** The results of fragment-constrained generation are shown in Table 2. GenMol significantly outperforms SAFE-GPT on most metrics across the tasks, demonstrating its general applicability to a variety of fragment constraint generation settings. Especially, GenMol generates high-quality molecules while preserving high diversity under the fragment constraints, again validating GenMol can strike an improved balance between quality and diversity.

## 5.3. Goal-directed Hit Generation

**Setup.** In goal-directed hit generation, the goal is to generate hits, i.e., molecules with optimized target chemical properties. Following Lee et al. (2024b) and Lee et al. (2024a), we construct an initial fragment vocabulary by decomposing the molecules in the ZINC250k dataset (Irwin et al., 2012). We adopt the practical molecular optimization (PMO) benchmark (Gao et al., 2022) which contains 23 tasks. The maximum number of oracle calls is set to 10,000 and performance is measured using the area under the curve (AUC) of the average top-10 property scores versus oracle calls. As our baselines, we adopt the recent state-of-the-art methods, $f$-RAG (Lee et al., 2024a), Genetic GFN (Kim et al., 2024), and Mol GA (Tripp & Hernández-Lobato, 2023). We also report the results of the top four methods in Gao et al. (2022). Note that since Gao et al. (2022) reported the results of a total of 25 methods, comparing GenMol to the top methods is equivalent to comparing it to 25 methods. Further details are provided in Section D.5.

**Results.** The results are shown in Table 3. As shown in the table, GenMol significantly outperforms the previous methods in terms of the sum AUC top-10 value and exhibits the best performance in 19 out of 23 tasks by a large margin. These results verify that the proposed optimization strategy of GenMol with fragment remasking is effective in discovering optimized hits. The results of additional baselines are provided in Table 13, Table 14, and Table 15.

## 5.4. Goal-directed Lead Optimization

**Setup.** Given an initial seed molecule, the goal in goal-directed lead optimization is to generate leads, i.e., molecules that exhibit improved target properties while maintaining the similarity with the given seed. Following Wang et al. (2023), the objective is to optimize the binding affinity to the target protein while satisfying the following constraints: QED $\geq$ 0.6, SA $\leq$ 4, and sim $\geq \delta$ where $\delta \in \{0.4, 0.6\}$ and sim is the Tanimoto similarity between the Morgan fingerprints of the generated molecules and the seed. Performance is evaluated by the docking score of the most optimized lead. Following Lee et al. (2023), we adopt five target proteins, **parp1**, **fa7**, **5ht1b**, **braf**, and **jak2**. For

Table 3: **Goal-directed hit generation results.** The results are the means of PMO AUC top-10 of 3 runs. The results for $f$-RAG (Lee et al., 2024a), Genetic GFN (Kim et al., 2024) and Mol GA (Tripp & Hernández-Lobato, 2023) are taken from the respective papers and the results for other baselines are taken from Gao et al. (2022). The best results are highlighted in bold.

| Oracle | GenMol | $f$-RAG | Genetic GFN | Mol GA | REINVENT | Graph GA | SELFIES-REINVENT | GP BO |
|---|---|---|---|---|---|---|---|---|
| albuterol_similarity | 0.937 | **0.977** | 0.949 | 0.896 | 0.882 | 0.838 | 0.826 | 0.898 |
| amlodipine_mpo | **0.810** | 0.749 | 0.761 | 0.688 | 0.635 | 0.661 | 0.607 | 0.583 |
| celecoxib_rediscovery | **0.826** | 0.778 | 0.802 | 0.567 | 0.713 | 0.630 | 0.573 | 0.723 |
| deco_hop | **0.960** | 0.936 | 0.733 | 0.649 | 0.666 | 0.619 | 0.631 | 0.629 |
| drd2 | **0.995** | 0.992 | 0.974 | 0.936 | 0.945 | 0.964 | 0.943 | 0.923 |
| fexofenadine_mpo | **0.894** | 0.856 | 0.856 | 0.825 | 0.784 | 0.760 | 0.741 | 0.722 |
| gsk3b | **0.986** | 0.969 | 0.881 | 0.843 | 0.865 | 0.788 | 0.780 | 0.851 |
| isomers_c7h8n2o2 | 0.942 | 0.955 | **0.969** | 0.878 | 0.852 | 0.862 | 0.849 | 0.680 |
| isomers_c9h10n2o2pf2cl | 0.833 | 0.850 | **0.897** | 0.865 | 0.642 | 0.719 | 0.733 | 0.469 |
| jnk3 | **0.906** | 0.904 | 0.764 | 0.702 | 0.783 | 0.553 | 0.631 | 0.564 |
| median1 | **0.398** | 0.340 | 0.379 | 0.257 | 0.356 | 0.294 | 0.355 | 0.301 |
| median2 | **0.359** | 0.323 | 0.294 | 0.301 | 0.276 | 0.273 | 0.255 | 0.297 |
| mestranol_similarity | **0.982** | 0.671 | 0.708 | 0.591 | 0.618 | 0.579 | 0.620 | 0.627 |
| osimertinib_mpo | **0.876** | 0.866 | 0.860 | 0.844 | 0.837 | 0.831 | 0.820 | 0.787 |
| perindopril_mpo | **0.718** | 0.681 | 0.595 | 0.547 | 0.537 | 0.538 | 0.517 | 0.493 |
| qed | **0.942** | 0.939 | **0.942** | 0.941 | 0.941 | 0.940 | 0.940 | 0.937 |
| ranolazine_mpo | **0.821** | 0.820 | 0.819 | 0.804 | 0.760 | 0.728 | 0.748 | 0.735 |
| scaffold_hop | **0.628** | 0.576 | 0.615 | 0.527 | 0.560 | 0.517 | 0.525 | 0.548 |
| sitagliptin_mpo | 0.584 | 0.601 | **0.634** | 0.582 | 0.021 | 0.433 | 0.194 | 0.186 |
| thiothixene_rediscovery | **0.692** | 0.584 | 0.583 | 0.519 | 0.534 | 0.479 | 0.495 | 0.559 |
| troglitazone_rediscovery | **0.867** | 0.448 | 0.511 | 0.427 | 0.441 | 0.390 | 0.348 | 0.410 |
| valsartan_smarts | **0.822** | 0.627 | 0.135 | 0.000 | 0.178 | 0.000 | 0.000 | 0.000 |
| zaleplon_mpo | **0.584** | 0.486 | 0.552 | 0.519 | 0.358 | 0.346 | 0.333 | 0.221 |
| Sum | **18.362** | 16.928 | 16.213 | 14.708 | 14.196 | 13.751 | 13.471 | 13.156 |

each target, three molecules from its known active ligands are selected and each is given as a seed molecule, yielding a total of 30 tasks. An initial fragment vocabulary is constructed by decomposing the seed molecule. If the generated molecule is lead, its fragments are added to the vocabulary. Following Wang et al. (2023), we adopt Graph GA (Jensen, 2019) and RetMol (Wang et al., 2023) as our baselines. Further details are provided in Section D.6.

**Results.** The results of goal-directed lead optimization are shown in Table 4. As shown in the table, GenMol outperforms the baselines in most tasks. Note that baselines often fail, i.e., they cannot generate molecules with a higher binding affinity than the seed molecule while satisfying the constraints, especially under the harsher ($\delta = 0.6$) similarity constraint. In contrast, GenMol is able to successfully optimize seed molecules in 26 out of 30 tasks, validating its effectiveness in exploring chemical space to optimize given molecules and discover promising lead molecules.

### 5.5. Ablation Study

**Fragment remasking.** To examine the effect of the proposed fragment remasking with masked discrete diffusion, we conduct ablation studies with alternative remasking strategies in Table 5. **Attaching (A)** is a baseline that attaches two fragments from the vocabulary without further modifications. On top of it, **A + Token remasking** randomly re-predicts individual tokens instead of a fragment chunk with discrete diffusion and **A + GPT remasking** re-predicts a randomly chosen fragment chunk with SAFE-

GPT instead of diffusion. **A + Fragment remasking (F)** re-predicts a fragment mask chunk with discrete diffusion. First, A + Token remasking, A + GPT remasking and A + Fragment remasking all outperform A, highlighting the importance of exploration through remasking. A + Fragment remasking outperforms A + Token remasking, proving that using fragments as the exploration unit is aligned with chemical intuition and effective in chemical exploration. A + Fragment remasking is also superior to A + GPT remasking, proving the effectiveness of the masked discrete diffusion that does not rely on specific ordering of tokens with bidirectional attention. We also conduct the ablation studies on lead optimization in Table 17. Although the naive baseline A outperforms other previous baselines in hit generation, it cannot generate new fragments outside of the vocabulary and therefore fails frequently in lead optimization, and applying fragment remasking on top of it largely improves lead optimization performance.

**Molecular context guidance.** To verify the effect of MCG, we present the results of GenMol with (**A + F + MCG**) and without (**A + F**; i.e., $\gamma = 0$) MCG in Table 5. A + F + MCG shows its superiority over A + F, demonstrating that calibrating GenMol's predictions with molecular context information with MCG improves GenMol's performance. The full results are shown in Table 16, where A + F + MCG achieves the best performance in 19 out of 23 tasks. The same trend is also observed in Table 12, where GenMol w/ MCG outperforms GenMol w/o MCG across various tasks on fragment-constrained generation.

Table 4: **Lead optimization results (kcal/mol).** The results are the mean docking scores of the most optimized leads of 3 runs. Lower is better.

| Target protein | Seed score | $\delta = 0.4$ | | | $\delta = 0.6$ | | |
|---|---|---|---|---|---|---|---|
| | | GenMol | RetMol | Graph GA | GenMol | RetMol | Graph GA |
| parp1 | -7.3 | **-10.6** | -9.0 | -8.3 | **-10.4** | - | -8.6 |
| | -7.8 | **-11.0** | -10.7 | -8.9 | **-9.7** | - | -8.1 |
| | -8.2 | **-11.3** | -10.9 | - | **-9.2** | - | - |
| fa7 | -6.4 | **-8.4** | -8.0 | -7.8 | -7.3 | **-7.6** | **-7.6** |
| | -6.7 | **-8.4** | - | -8.2 | **-7.6** | - | **-7.6** |
| | -8.5 | - | - | - | - | - | - |
| 5ht1b | -4.5 | **-12.9** | -12.1 | -11.7 | **-12.1** | - | -11.3 |
| | -7.6 | **-12.3** | -9.0 | -12.1 | **-12.0** | -10.0 | **-12.0** |
| | -9.8 | **-11.6** | - | - | **-10.5** | - | - |
| braf | -9.3 | **-10.8** | - | -9.8 | - | - | - |
| | -9.4 | -10.8 | **-11.6** | - | **-9.7** | - | - |
| | -9.8 | -10.6 | - | **-11.6** | **-10.5** | - | -10.4 |
| jak2 | -7.7 | **-10.2** | -8.2 | -8.7 | **-9.3** | - | -8.1 |
| | -8.0 | **-10.0** | -9.0 | -9.2 | **-9.4** | - | -9.2 |
| | -8.6 | **-9.8** | - | - | - | - | - |

Table 5: **Ablation study** on the goal-directed hit generation task. The results are the mean sums of PMO AUC top-10 of 3 runs. The best results are highlighted in bold. The full results are shown in Table 16.

| Method | Sum |
|---|---|
| Attaching (A) | 17.641 |
| A + Token remasking | 18.091 (A+0.450) |
| A + GPT remasking | 18.074 (A+0.433) |
| A + Fragment remasking (F) | 18.208 (A+0.567) |
| A + F + MCG | **18.362** (A+F+0.154) |

# 6. Conclusion

We proposed GenMol, a molecule generation framework designed to deal with various drug discovery scenarios effectively and efficiently by integrating discrete diffusion with SAFE. Especially, fragment remasking allows GenMol to effectively explore chemical space and MCG further improves GenMol's performance. The experimental results showed that GenMol can achieve state-of-the-art results in a wide range of drug discovery tasks, demonstrating its potential as a unified and versatile tool for drug discovery.

# Impact Statement

In our paper, we showed that GenMol is capable of addressing a broad spectrum of drug discovery challenges, providing a unified and versatile solution for molecular design. However, as effective as GenMol is in drug discovery tasks, it has the potential to generate harmful drugs if used maliciously. To prevent this, GenMol could be equipped with features that incorporate target properties that take toxicity into account, exclude toxic fragments from the fragment vocabulary, or filter the proposed drug candidates by predicting the toxicity.

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

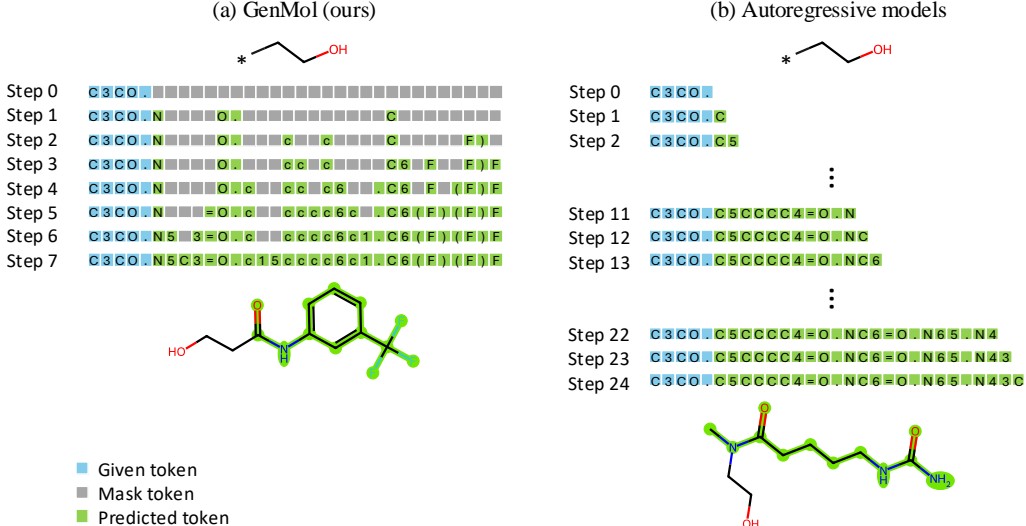

Figure 5: **Illustration of the fragment-constrained motif extension process of (a) GenMol and (b) autoregressive models.** GenMol starts by sampling the length of the sequence and then filling the sequence with mask tokens corresponding to the sampled length. GenMol employs parallel decoding where all tokens are decoded simultaneously under discrete diffusion, and confirms only the most confident predictions. The decoding proceeds progressively until all mask tokens are predicted. In contrast, autoregressive models like SAFE-GPT (Noutahi et al., 2024) need to predict one token per step, requiring many more decoding steps.

## A. Limitations

The proposed GenMol is designed with the goal of being a versatile tool for various drug discovery scenarios without any task-specific finetuning. In addition to its versatility, GenMol shows improved molecular generation in terms of both effectiveness and efficiency thanks to its masked discrete diffusion with non-autoregressive bidirectional parallel decoding. However, while the parallel decoding scheme reduces sampling time, unmasking more than one token at each time step (i.e., setting $N > 1$) results in degraded generation quality. Overcoming this trade-off between generation quality and sampling efficiency is left as a future work.

## B. Confidence Sampling of GenMol

Following Chang et al. (2022), GenMol adopts confidence sampling that decides which tokens to unmask at each sampling step based on the confidence scores of the sampled tokens. After sampling $z_s^l$ according to Eq. (4), let $p_{\theta,i*}^l$ denote the corresponding prediction score, where $i^*$ is the index of the sampled category. With introduction of a Gumbel noise decreasing over the sampling process, the confidence score $c_t^l$ of the $l$-th token at time step $t$ is defined as follows:

$$c_t^l := \log p_{\theta,i*}^l + r \cdot t \cdot \epsilon, \quad \epsilon \sim \text{Gumbel}(0, 1), \tag{8}$$

where $r$ is the randomness.

Based on their confidence scores $c_t^l$ (Eq. (8)), GenMol unmask the top-$N$ currently masked tokens. In other words, this is equivalent to predicting all tokens simultaneously, but only confirming the most confident predictions. The other tokens are masked again and predicted in the next step. With confidence sampling, GenMol can exploit the dependencies between tokens in a given molecular sequence for better sampling quality, rather than randomly and independently selecting tokens to unmask, as in the standard diffusion sampling (Austin et al., 2021; Sahoo et al., 2024).

# C. Derivation of MCG (Eq. (7))

Nisonoff et al. (2024) proposed a CFG scheme for CTMC-based discrete diffusion models (Campbell et al., 2022) as follows:

$$R_t^{(w)}(\boldsymbol{z}_t, \boldsymbol{z}_s | \boldsymbol{y}) = R_t(\boldsymbol{z}_t, \boldsymbol{z}_s | \boldsymbol{y})^w R_t(\boldsymbol{z}_t, \boldsymbol{z}_s)^{1-w}, \tag{9}$$

where $R(\boldsymbol{z}, \boldsymbol{z}')$ is a rate matrix that describes the transition probability for $\boldsymbol{z} \to \boldsymbol{z}'$.

Applying the same generalization of CFG proposed by Karras et al. (2024) (Eq. (6)) to Eq. (9), we have the following guided rate matrix:

$$R_t^{(w)}(\boldsymbol{z}_t, \boldsymbol{z}_s | \boldsymbol{y}) = R_{1,t}(\boldsymbol{z}_t, \boldsymbol{z}_s | \boldsymbol{y})^w R_{0,t}(\boldsymbol{z}_t, \boldsymbol{z}_s | \boldsymbol{y})^{1-w}, \tag{10}$$

where $R_{1,t}$ and $R_{0,t}$ are the rate matrices describing a high-quality model and a poor model, respectively.

On the other hand, MDLM can be interpreted as a CTMC with the following reverse rate matrix (Sahoo et al., 2024):

$$R_t(\boldsymbol{z}_t, \boldsymbol{z}_s) = \begin{cases} -\frac{\alpha'_t}{1-\alpha_t} \langle \boldsymbol{x}_\theta(\boldsymbol{z}_t, t), \boldsymbol{z}_s \rangle & \boldsymbol{z}_s \neq \mathbf{m}, \boldsymbol{z}_t = \mathbf{m} \\ \frac{\alpha'_t}{1-\alpha_t} & \boldsymbol{z}_s = \mathbf{m}, \boldsymbol{z}_t = \mathbf{m} \\ 0 & \text{otherwise.} \end{cases} \tag{11}$$

If we set $R_{1,t}$ and $R_{0,t}$ as the rate matrices resulted by the original prediction of the denoiser $\boldsymbol{x}_\theta(\boldsymbol{z}_t, t)$ and the prediction with partially corrupted input $\boldsymbol{x}_\theta(\tilde{\boldsymbol{z}}_t, t)$, respectively, substituting Eq. (11) for $R_{1,t}$ and $R_{0,t}$ in Eq. (10) yields:

$$\begin{aligned} R_t^{(w)}(\boldsymbol{z}_t, \boldsymbol{z}_s) &= \begin{cases} -\frac{\alpha'_t}{1-\alpha_t} \langle \boldsymbol{x}_\theta(\boldsymbol{z}_t, t), \boldsymbol{z}_s \rangle^w \langle \boldsymbol{x}_\theta(\tilde{\boldsymbol{z}}_t, t), \boldsymbol{z}_s \rangle^{1-w} & \boldsymbol{z}_s \neq \mathbf{m}, \boldsymbol{z}_t = \mathbf{m} \\ \frac{\alpha'_t}{1-\alpha_t} & \boldsymbol{z}_s = \mathbf{m}, \boldsymbol{z}_t = \mathbf{m} \\ 0 & \text{otherwise} \end{cases} \\ &= \begin{cases} -\frac{\alpha'_t}{1-\alpha_t} \langle \boldsymbol{x}_\theta(\boldsymbol{z}_t, t)^w \odot \boldsymbol{x}_\theta(\tilde{\boldsymbol{z}}_t, t)^{1-w}, \boldsymbol{z}_s \rangle & \boldsymbol{z}_s \neq \mathbf{m}, \boldsymbol{z}_t = \mathbf{m} \\ \frac{\alpha'_t}{1-\alpha_t} & \boldsymbol{z}_s = \mathbf{m}, \boldsymbol{z}_t = \mathbf{m} \\ 0 & \text{otherwise,} \end{cases} \end{aligned} \tag{12}$$

where $\odot$ denotes the Hadamard product. Here, we utilized the facts that $\boldsymbol{z}_s$ is an one-hot vector and $\boldsymbol{z}_t = \mathbf{m} \Rightarrow \tilde{\boldsymbol{z}}_t = \mathbf{m}$ as $\tilde{\boldsymbol{z}}_t$ is the corrupted (i.e., more masked) $\boldsymbol{z}_t$.

Therefore, using the guided rate matrix $R_t^{(w)}$ in Eq. (12) is equivalent to using the following guided prediction $\boldsymbol{x}_\theta^{(w)}$:

$$\begin{aligned} \boldsymbol{x}_\theta^{(w)}(\boldsymbol{z}_t, t) &= \boldsymbol{x}_\theta(\boldsymbol{z}_t, t)^w \odot \boldsymbol{x}_\theta(\tilde{\boldsymbol{z}}_t, t)^{1-w} \\ \Leftrightarrow \ \log \boldsymbol{x}_\theta^{(w)}(\boldsymbol{z}_t, t) &= w \log \boldsymbol{x}_\theta(\boldsymbol{z}_t, t) + (1-w) \log \boldsymbol{x}_\theta(\tilde{\boldsymbol{z}}_t, t). \end{aligned} \tag{13}$$

Table 6: **Statistics of the SAFE dataset.**

|  | Train | Test | Validation |
|---|---|---|---|
| Number of examples | 945,455,307 | 118,890,444 | 118,451,032 |

## D. Experimental Details

### D.1. Computing Resources

GenMol was trained using 8 NVIDIA A100 GPUs. The training took approximately 5 hours. All the molecular generation experiments were conducted using a single NVIDIA A100 GPU and 32 CPU cores.

### D.2. Training GenMol

In this section, we describe the details for training GenMol. We used the BERT (Devlin et al., 2019) architecture of the HuggingFace Transformers library (Wolf et al., 2019) with the default configuration, except that we set max_position_embeddings to 256. We used the SAFE dataset and SAFE tokenizer (Noutahi et al., 2024) that has a vocabulary size of $K = 1880$. The statistics of the dataset is provided in Table 6. We set the batch size to 2048, the learning rate to $3e - 4$, and the number of training steps to 50k. We used the log-linear noise schedule of Sahoo et al. (2024) and the AdamW optimizer (Loshchilov & Hutter, 2019) with $\beta_1 = 0.9$ and $\beta_2 = 0.999$.

### D.3. *De Novo* Generation

In this section, we describe the details for conducting experiments in Section 5.1. We used the RDKit library (Landrum et al., 2016) to obtain Morgan fingerprints and the Therapeutics Data Commons (TDC) library (Huang et al., 2021) to calculate diversity, QED, and SA. The lengths of the mask chunks were sampled from the ZINC250k distribution.

### D.4. Fragment-constrained Generation

In Section 5.2, we used the benchmark proposed by Noutahi et al. (2024). The benchmark contains extracted fragments from 10 known drugs: Cyclothiazide, Maribavir, Spirapril, Baricitinib, Eliglustat, Erlotinib, Futibatinib, Lesinurad, Liothyronine, and Lovastatin. Specifically, from each drug, side chains, a starting motif, the main scaffold with attachment points, and a core substructure are extracted, and then serve as input for linker design & scaffold morphing, motif extension, scaffold decoration, and superstructure generation, respectively. **Linker design** and **scaffold morphing** are tasks where the goal is to generate a linker fragment that connects given two side chains. In GenMol, linker design and scaffold morphing correspond to the same task. **Motif extension** and **scaffold decoration** are tasks where the goal is to generate a side fragment to complete a new molecule when a motif or scaffold and attachment points are given. **Superstructure generation** is a task where the goal is to generate a molecule when a substructure constraint is given. Following Noutahi et al. (2024), we first generate random attachment points on the substructure to create new scaffolds and conduct the scaffold decoration task.

We used $N = 1$. We performed the grid search with the search space $\tau \in \{0.5, 0.8, 1, 1.2, 1.5\}$ and $r \in \{1, 1.2, 2, 3\}$, and set $r$ to 3 for linker design and scaffold morphing, 1.2 for motif extension, and 2 for scaffold decoration and superstructure generation. We set $\tau$ to 1.2 for all the tasks. The lengths of the mask chunks were sampled from the ZINC250k distribution. For MCG, we set $w = 2$ and performed a search with the search space $\gamma \in \{0, 0.1, 0.2, 0.3, 0.4, 0.5\}$. The values of $\gamma$ are provided in Table 7.

### D.5. Goal-directed Hit Generation

In this section, we describe the details for conducting experiments in Section 5.3. To construct an initial fragment vocabulary, we adopted a simple decomposition rule $R_{\text{vocab}}$ that randomly cut one of the non-ring single bonds of a given molecule three times and apply it to the ZINC250k dataset. With this decomposition rule, we can ensure that all fragments have one attachment point and are of appropriate size. For the finer decomposition rule $R_{\text{remask}}$ that determines which fragments fragment remasking will operate on, we used the rule that cut all of the non-ring single bonds in a given molecule. We set the size of the fragment vocabulary to $V = 100$. We applied the warmup scheme that let GenMol generate molecules by concatenating two randomly chosen fragments without fragment remasking for the first 1,000 generations. We used $N = 1$,

Table 8: $\gamma$ **in hit generation.**

| Oracle | $\gamma$ |
|---|---|
| albuterol_similarity | 0.2 |
| amlodipine_mpo | 0.3 |
| celecoxib_rediscovery | 0.0 |
| deco_hop | 0.2 |
| drd2 | 0.0 |
| fexofenadine_mpo | 0.0 |
| gsk3b | 0.0 |
| isomers_c7h8n2o2 | 0.5 |
| isomers_c9h10n2o2pf2cl | 0.0 |
| jnk3 | 0.5 |
| median1 | 0.2 |
| median2 | 0.2 |
| mestranol_similarity | 0.0 |
| osimertinib_mpo | 0.0 |
| perindopril_mpo | 0.4 |
| qed | 0.0 |
| ranolazine_mpo | 0.0 |
| scaffold_hop | 0.0 |
| sitagliptin_mpo | 0.2 |
| thiothixene_rediscovery | 0.3 |
| troglitazone_rediscovery | 0.0 |
| valsartan_smarts | 0.4 |
| zaleplon_mpo | 0.4 |

Table 7: $\gamma$ **in fragment-constrained generation.**

| Task | $\gamma$ |
|---|---|
| Linker design | 0.0 |
| Scaffold morphing | 0.0 |
| Motif extension | 0.3 |
| Scaffold decoration | 0.3 |
| Superstructure generation | 0.4 |

Table 9: $\gamma$ **in lead optimization.**

| Target protein | Seed score | $\gamma$ |
|---|---|---|
| parp1 | -7.3 | 0.2 |
| | -7.8 | 0.2 |
| | -8.2 | 0.2 |
| fa7 | -6.4 | 0.3 |
| | -6.7 | 0.4 |
| | -8.5 | 0.0 |
| 5ht1b | -4.5 | 0.3 |
| | -7.6 | 0.0 |
| | -9.8 | 0.4 |
| braf | -9.3 | 0.2 |
| | -9.4 | 0.1 |
| | -9.8 | 0.5 |
| jak2 | -7.7 | 0.5 |
| | -8.0 | 0.0 |
| | -8.6 | 0.1 |

$\tau = 1.2$, and $r = 2$. For MCG, we set $w = 2$ and performed a search with the search space $\gamma \in \{0, 0.1, 0.2, 0.3, 0.4, 0.5\}$. The values of $\gamma$ are shown in Table 8.

## D.6. Goal-directed Lead Optimization

In this section, we describe the details for conducting experiments in Section 5.4. For each target, three molecules were randomly selected from known active compounds from DUD-E (Mysinger et al., 2012) (parp1, fa7, braf and jak2) or ChEMBL (Zdrazil et al., 2024) (5ht1b) as seed molecules. The same decomposition rules $R_{\text{vocab}}$ and $R_{\text{remask}}$ explained in Section D.5 were used and the fragment vocabulary size was set to $V = \infty$. Following the setting of Wang et al. (2023), for each target protein and each seed molecule, we run 10 optimization iterations with 100 generation per iteration. We used $N = 1$, $\tau = 1.2$, and $r = 2$. For MCG, we set $w = 2$ and performed a search with the search space $\gamma \in \{0, 0.1, 0.2, 0.3, 0.4, 0.5\}$. The values of $\gamma$ are shown in Table 9.

# E. Additional Experimental Results

## E.1. Additional Baseline in *De Novo* Generation

We provide comparison of GenMol with another widely used baseline trained on ZINC250k (Irwin et al., 2012), JT-VAE (Jin et al., 2018), and a graph discrete diffusion model trained on GuacaMol (Brown et al., 2019), DiGress (Vignac et al., 2023), in Table 10. GenMol significantly outperforms JT-VAE and DiGress in terms of quality and sampling time. We used the fast version of the JT-VAE code[1] and the official code of DiGress[2].

## E.2. QED and SA Distributions in *De Novo* Generation

We provide the QED and SA distributions of molecules generated by GenMol and SAFE-GPT, respectively, in Figure 6. The distributions of 100k molecules randomly sampled from the test set are also shown in the figure. As shown in the figure, GenMol is able to generate molecules of higher QED (more drug-like) and lower SA (more synthesizable) values

---

[1] https://github.com/Bibyutatsu/FastJTNNpy3
[2] https://github.com/cvignac/DiGress

Table 10: *De novo* **molecule generation results.** The results are the means and the standard deviations of 3 runs. $N$, $\tau$, and $r$ is the number of tokens to unmask at each time step, the softmax temperature, and the randomness, respectively. The best results are highlighted in bold.

| Method | Validity (%) | Uniqueness (%) | Quality (%) | Diversity | Sampling time (s) |
|---|---|---|---|---|---|
| JT-VAE | **100.0** $\pm$ 0.0 | 65.9 $\pm$ 1.3 | 45.2 $\pm$ 1.4 | 0.855 $\pm$ 0.001 | 96.5 $\pm$ 2.1 |
| DiGress | 89.6 $\pm$ 0.8 | **100.0** $\pm$ 0.0 | 36.8 $\pm$ 1.0 | **0.885** $\pm$ 0.002 | 1241.9 $\pm$ 9.2 |
| GenMol ($N = 1$) | | | | | |
| $\quad \tau = 0.5, r = 0.5$ | **100.0** $\pm$ 0.0 | 99.7 $\pm$ 0.1 | **84.6** $\pm$ 0.8 | 0.818 $\pm$ 0.001 | **21.1** $\pm$ 0.4 |
| $\quad \tau = 1.0, r = 10.0$ | 99.8 $\pm$ 0.1 | 99.6 $\pm$ 0.1 | 63.0 $\pm$ 0.4 | 0.882 $\pm$ 0.003 | 21.5 $\pm$ 0.5 |

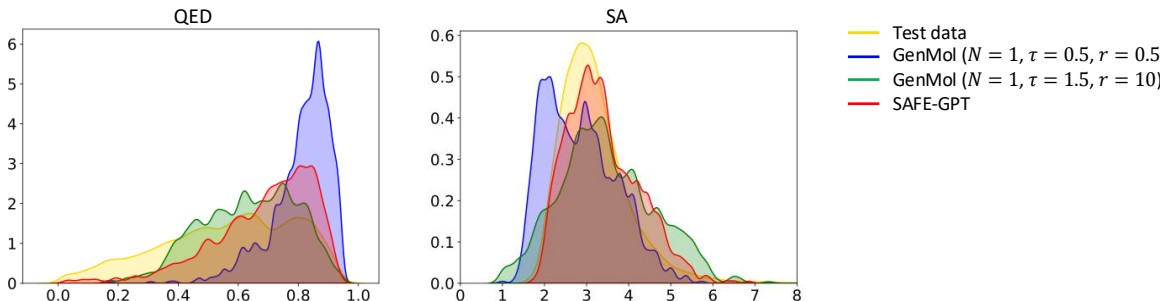

Figure 6: **QED and SA distributions of molecules in *de novo* generation.**

Table 11: **Quality (%) results in *de novo* molecule generation** with various QED and SA thresholds. The results are the means and the standard deviations of 3 runs. The best results are highlighted in bold.

| Method | Quality | | |
|---|---|---|---|
| | QED $\geq 0.6$, SA $\leq 4$ | QED $\geq 0.5$, SA $\leq 5$ | QED $\geq 0.7$, SA $\leq 3$ |
| SAFE-GPT | 54.7 $\pm$ 0.3 | 78.3 $\pm$ 1.7 | 18.5 $\pm$ 0.1 |
| GenMol ($N = 1, \tau = 0.5, r = 0.5$) | **84.6** $\pm$ 0.8 | **97.1** $\pm$ 0.1 | **50.6** $\pm$ 1.0 |

than SAFE-GPT, resulting in high quality in Table 1. Furthermore, GenMol can freely control these distributions by adjusting the values of the softmax temperature $\tau$ and the randomness $r$.

### E.3. Analysis on Quality and Diversity in *De Novo* Generation

The quality and diversity values of 100k molecules randomly sampled from the test set are 38.2% and 0.897, respectively. The quality and diversity values of molecules generated by GenMol ($N = 1$, $\tau = 0.5$, $r = 0.5$) are 84.6% and 0.818, respectively, as shown in Table 1. We can think of this as sacrificing diversity and selecting a specific mode of high quality. GenMol can control this mode-selecting behavior by adjusting $\tau$ and $r$ (e.g., the quality and diversity values of GenMol ($N = 1$, $\tau = 1.5$, $r = 10$) shown in Table 1 are 39.7% and 0.911, respectively).

### E.4. Analysis on QED and SA Thresholds in Quality Metric

In Section 5.1 and Section 5.2, we introduced quality to provide a metric that summarizes the model's ability to generate chemically plausible molecules. We followed Jin et al. (2020) and set the QED and SA thresholds to 0.6 and 4, respectively. To validate the robustness of the quality metric to threshold selection, we additionally provide quality results on the *de novo* generation task with different thresholds in Table 11. QED $\geq 0.5$, SA $\leq 5$ corresponds to the softer condition, while QED $\geq 0.7$, SA $\leq 3$ is the harsher condition. As shown in the table, different QED and SA thresholds yield a consistent quality trend, with GenMol outperforming SAFE-GPT in all three settings.

### E.5. Additional Baseline in Fragment-constrained Generation

We compare the results of GenMol and DiGress in the fragment-constrained generation tasks in Table 12. As shown in the table, GenMol significantly outperforms DiGress in validity and quality.

Table 12: **Fragment-constrained molecule generation results.** The results are the means and the standard deviations of 3 runs. The purple parentheses indicate the performance gain by MCG.

| Method | Task | Validity (%) | Uniqueness (%) | Quality (%) | Diversity | Distance |
|---|---|---|---|---|---|---|
| DiGress | Linker design | $31.2 \pm 1.2$ | $84.3 \pm 0.4$ | $6.1 \pm 0.2$ | $0.745 \pm 0.001$ | $0.724 \pm 0.003$ |
| | Scaffold morphing | $31.2 \pm 1.2$ | $84.3 \pm 0.4$ | $6.1 \pm 0.2$ | $0.745 \pm 0.001$ | $0.724 \pm 0.003$ |
| | Motif extension | $21.8 \pm 0.8$ | $94.5 \pm 0.3$ | $4.2 \pm 0.1$ | $0.818 \pm 0.003$ | $0.794 \pm 0.003$ |
| | Scaffold decoration | $29.3 \pm 0.7$ | $91.0 \pm 0.8$ | $9.1 \pm 0.4$ | $0.793 \pm 0.003$ | $0.785 \pm 0.002$ |
| | Superstructure generation | $26.7 \pm 1.3$ | $85.9 \pm 1.4$ | $7.4 \pm 0.9$ | $0.789 \pm 0.005$ | $0.776 \pm 0.004$ |
| GenMol w/o MCG | Linker design | $100.0 \pm 0.0$ | $83.7 \pm 0.5$ | $21.9 \pm 0.4$ | $0.547 \pm 0.002$ | $0.563 \pm 0.003$ |
| | Scaffold morphing | $100.0 \pm 0.0$ | $83.7 \pm 0.5$ | $21.9 \pm 0.4$ | $0.547 \pm 0.002$ | $0.563 \pm 0.003$ |
| | Motif extension | $77.2 \pm 0.1$ | $77.8 \pm 0.2$ | $27.5 \pm 0.8$ | $0.617 \pm 0.002$ | $0.682 \pm 0.002$ |
| | Scaffold decoration | $96.8 \pm 0.2$ | $78.0 \pm 1.2$ | $29.6 \pm 0.8$ | $0.576 \pm 0.001$ | $0.650 \pm 0.001$ |
| | Superstructure generation | $98.2 \pm 1.1$ | $78.3 \pm 3.4$ | $33.3 \pm 1.6$ | $0.574 \pm 0.008$ | $0.757 \pm 0.003$ |
| GenMol | Linker design | $100.0 \pm 0.0$ (+0.0) | $83.7 \pm 0.5$ (+0.0) | $21.9 \pm 0.4$ (+0.0) | $0.547 \pm 0.002$ (+0.000) | $0.563 \pm 0.003$ (+0.000) |
| | Scaffold morphing | $100.0 \pm 0.0$ (+0.0) | $83.7 \pm 0.5$ (+0.0) | $21.9 \pm 0.4$ (+0.0) | $0.547 \pm 0.002$ (+0.000) | $0.563 \pm 0.003$ (+0.000) |
| | Motif extension | $82.9 \pm 0.1$ (+5.7) | $77.5 \pm 0.1$ (-0.3) | $30.1 \pm 0.4$ (+2.6) | $0.617 \pm 0.002$ (+0.000) | $0.682 \pm 0.001$ (+0.000) |
| | Scaffold decoration | $96.6 \pm 0.8$ (-0.2) | $82.7 \pm 1.8$ (+4.7) | $31.8 \pm 0.5$ (+2.2) | $0.591 \pm 0.001$ (+0.015) | $0.651 \pm 0.001$ (+0.001) |
| | Superstructure generation | $97.5 \pm 0.9$ (-0.7) | $83.6 \pm 1.0$ (+5.3) | $34.8 \pm 1.0$ (+1.5) | $0.599 \pm 0.009$ (+0.025) | $0.762 \pm 0.007$ (+0.005) |

### E.6. Ablation Study on MCG in Fragment-constrained Generation

We compare the results of GenMol and GenMol without MCG in the fragment-constrained generation tasks in Table 12. As shown in the table, using MCG improves the performance of GenMol across various tasks and metrics, verifying the effectiveness of the proposed MCG scheme.

### E.7. Full Goal-directed Hit Generation Results

We provide the full results of Table 3 including the additional baselines from Gao et al. (2022) in Table 13, Table 14, and Table 15. As shown in the tables, GenMol outperforms all baselines by a large margin.

On the other hand, to simulate the fragment-based drug discovery (FBDD) scenario following Lee et al. (2024a), we have assumed that a high-quality fragment vocabulary of size 100 is given for each task in Table 3. Here, similar to Wang et al. (2023), we also provide the results of GenMol (10 fragments), which starts with only 10 of these fragments to simulate a scenario where high-quality fragments are sparse, in Table 13.

### E.8. Full Goal-directed Hit Generation Results in Ablation Study

We provide the full results of the ablated GenMol variants baselines in the goal-directed hit generatino task in Table 5 in Table 16. As shown in the table, A, A + Token remasking, A + GPT remasking and A + Fragment remasking show inferior performance to A + F + MCG, the full GenMol, as discussed in Section 5.5.

### E.9. Ablation Study in Goal-directed Lead Optimization

We compare the results of the ablated GenMol baselines in the goal-directed lead optimization task in Table 17. As in Table 5, A + Token remasking, A + GPT remasking and A + Fragment remasking all outperform A, demonstrating the importance of further modification on top of A through remasking. Moreover, A + Fragment remasking generally outperforms A + Token remasking and A + GPT remasking, showing the effectiveness of the proposed remasking strategy that uses fragments as the exploration unit under discrete diffusion. Lastly, A + F + MCG shows further improved performance compared to A + F, validating the effectiveness of MCG in the lead optimization task.

### E.10. Examples of Generated Molecules

We provide examples of the molecules generated by GenMol ($N = 1$, $\tau = 0.5$, $r = 0.5$) on *de novo* generation Figure 7. We provide examples of generated molecules on fragment-constrained generation in Figure 8.

Table 13: **Goal-directed hit generation results.** The results are the means and standard deviations of PMO AUC top-10 of 3 runs. The results for $f$-RAG (Lee et al., 2024a), Genetic GFN (Kim et al., 2024) and Mol GA (Tripp & Hernández-Lobato, 2023) are taken from the respective papers and the results for other baselines are taken from Gao et al. (2022). The best results are highlighted in bold.

| Oracle | GenMol | GenMol (10 fragments) | $f$-RAG | Genetic GFN | Mol GA |
|---|---|---|---|---|---|
| albuterol_similarity | 0.937 ± 0.010 | 0.847 ± 0.036 | **0.977** ± 0.002 | 0.949 ± 0.010 | 0.896 ± 0.035 |
| amlodipine_mpo | **0.810** ± 0.012 | 0.762 ± 0.012 | 0.749 ± 0.019 | 0.761 ± 0.019 | 0.688 ± 0.039 |
| celecoxib_rediscovery | **0.826** ± 0.018 | 0.619 ± 0.005 | 0.778 ± 0.007 | 0.802 ± 0.029 | 0.567 ± 0.083 |
| deco_hop | **0.960** ± 0.010 | 0.957 ± 0.016 | 0.936 ± 0.011 | 0.733 ± 0.109 | 0.649 ± 0.025 |
| drd2 | **0.995** ± 0.000 | **0.995** ± 0.000 | 0.992 ± 0.000 | 0.974 ± 0.006 | 0.936 ± 0.016 |
| fexofenadine_mpo | **0.894** ± 0.028 | 0.806 ± 0.008 | 0.856 ± 0.016 | 0.856 ± 0.039 | 0.825 ± 0.019 |
| gsk3b | **0.986** ± 0.003 | 0.985 ± 0.004 | 0.969 ± 0.003 | 0.881 ± 0.042 | 0.843 ± 0.039 |
| isomers_c7h8n2o2 | 0.942 ± 0.004 | **0.984** ± 0.002 | 0.955 ± 0.008 | 0.969 ± 0.003 | 0.878 ± 0.026 |
| isomers_c9h10n2o2pf2cl | 0.833 ± 0.014 | 0.866 ± 0.010 | 0.850 ± 0.005 | **0.897** ± 0.007 | 0.865 ± 0.012 |
| jnk3 | **0.906** ± 0.023 | 0.828 ± 0.007 | 0.904 ± 0.004 | 0.764 ± 0.069 | 0.702 ± 0.123 |
| median1 | **0.398** ± 0.000 | 0.336 ± 0.008 | 0.340 ± 0.007 | 0.379 ± 0.010 | 0.257 ± 0.009 |
| median2 | **0.359** ± 0.004 | 0.354 ± 0.000 | 0.323 ± 0.005 | 0.294 ± 0.007 | 0.301 ± 0.021 |
| mestranol_similarity | 0.982 ± 0.000 | **0.991** ± 0.002 | 0.671 ± 0.021 | 0.708 ± 0.057 | 0.591 ± 0.053 |
| osimertinib_mpo | **0.876** ± 0.008 | 0.870 ± 0.004 | 0.866 ± 0.009 | 0.860 ± 0.008 | 0.844 ± 0.015 |
| perindopril_mpo | **0.718** ± 0.012 | 0.695 ± 0.004 | 0.681 ± 0.017 | 0.595 ± 0.014 | 0.547 ± 0.022 |
| qed | 0.942 ± 0.000 | **0.943** ± 0.000 | 0.939 ± 0.001 | **0.942** ± 0.000 | 0.941 ± 0.001 |
| ranolazine_mpo | **0.821** ± 0.011 | 0.777 ± 0.016 | 0.820 ± 0.016 | 0.819 ± 0.018 | 0.804 ± 0.011 |
| scaffold_hop | 0.628 ± 0.008 | **0.648** ± 0.005 | 0.576 ± 0.014 | 0.615 ± 0.100 | 0.527 ± 0.025 |
| sitagliptin_mpo | 0.584 ± 0.034 | 0.588 ± 0.064 | 0.601 ± 0.011 | **0.634** ± 0.039 | 0.582 ± 0.040 |
| thiothixene_rediscovery | **0.692** ± 0.123 | 0.569 ± 0.013 | 0.584 ± 0.009 | 0.583 ± 0.034 | 0.519 ± 0.041 |
| troglitazone_rediscovery | **0.867** ± 0.022 | 0.848 ± 0.040 | 0.448 ± 0.017 | 0.511 ± 0.054 | 0.427 ± 0.031 |
| valsartan_smarts | **0.822** ± 0.042 | 0.803 ± 0.011 | 0.627 ± 0.058 | 0.135 ± 0.271 | 0.000 ± 0.000 |
| zaleplon_mpo | **0.584** ± 0.011 | 0.571 ± 0.016 | 0.486 ± 0.004 | 0.552 ± 0.033 | 0.519 ± 0.029 |
| Sum | **18.362** | 17.643 | 16.928 | 16.213 | 14.708 |

| Oracle | REINVENT | Graph GA | SELFIES-REINVENT | GP BO | STONED |
|---|---|---|---|---|---|
| albuterol_similarity | 0.882 ± 0.006 | 0.838 ± 0.016 | 0.826 ± 0.030 | 0.898 ± 0.014 | 0.745 ± 0.076 |
| amlodipine_mpo | 0.635 ± 0.035 | 0.661 ± 0.020 | 0.607 ± 0.014 | 0.583 ± 0.044 | 0.608 ± 0.046 |
| celecoxib_rediscovery | 0.713 ± 0.067 | 0.630 ± 0.097 | 0.573 ± 0.043 | 0.723 ± 0.053 | 0.382 ± 0.041 |
| deco_hop | 0.666 ± 0.044 | 0.619 ± 0.004 | 0.631 ± 0.012 | 0.629 ± 0.018 | 0.611 ± 0.008 |
| drd2 | 0.945 ± 0.007 | 0.964 ± 0.012 | 0.943 ± 0.005 | 0.923 ± 0.017 | 0.913 ± 0.020 |
| fexofenadine_mpo | 0.784 ± 0.006 | 0.760 ± 0.011 | 0.741 ± 0.002 | 0.722 ± 0.005 | 0.797 ± 0.016 |
| gsk3b | 0.865 ± 0.043 | 0.788 ± 0.070 | 0.780 ± 0.037 | 0.851 ± 0.041 | 0.668 ± 0.049 |
| isomers_c7h8n2o2 | 0.852 ± 0.036 | 0.862 ± 0.065 | 0.849 ± 0.034 | 0.680 ± 0.117 | 0.899 ± 0.011 |
| isomers_c9h10n2o2pf2cl | 0.642 ± 0.054 | 0.719 ± 0.047 | 0.733 ± 0.029 | 0.469 ± 0.180 | 0.805 ± 0.031 |
| jnk3 | 0.783 ± 0.023 | 0.553 ± 0.136 | 0.631 ± 0.064 | 0.564 ± 0.155 | 0.523 ± 0.092 |
| median1 | 0.356 ± 0.009 | 0.294 ± 0.021 | 0.355 ± 0.011 | 0.301 ± 0.014 | 0.266 ± 0.016 |
| median2 | 0.276 ± 0.008 | 0.273 ± 0.009 | 0.255 ± 0.005 | 0.297 ± 0.009 | 0.245 ± 0.032 |
| mestranol_similarity | 0.618 ± 0.048 | 0.579 ± 0.022 | 0.620 ± 0.029 | 0.627 ± 0.089 | 0.609 ± 0.101 |
| osimertinib_mpo | 0.837 ± 0.009 | 0.831 ± 0.005 | 0.820 ± 0.003 | 0.787 ± 0.006 | 0.822 ± 0.012 |
| perindopril_mpo | 0.537 ± 0.016 | 0.538 ± 0.009 | 0.517 ± 0.021 | 0.493 ± 0.011 | 0.488 ± 0.011 |
| qed | 0.941 ± 0.000 | 0.940 ± 0.000 | 0.940 ± 0.000 | 0.937 ± 0.000 | 0.941 ± 0.000 |
| ranolazine_mpo | 0.760 ± 0.009 | 0.728 ± 0.012 | 0.748 ± 0.018 | 0.735 ± 0.013 | 0.765 ± 0.029 |
| scaffold_hop | 0.560 ± 0.019 | 0.517 ± 0.007 | 0.525 ± 0.013 | 0.548 ± 0.019 | 0.521 ± 0.034 |
| sitagliptin_mpo | 0.021 ± 0.003 | 0.433 ± 0.075 | 0.194 ± 0.121 | 0.186 ± 0.055 | 0.393 ± 0.083 |
| thiothixene_rediscovery | 0.534 ± 0.013 | 0.479 ± 0.025 | 0.495 ± 0.040 | 0.559 ± 0.027 | 0.367 ± 0.027 |
| troglitazone_rediscovery | 0.441 ± 0.032 | 0.390 ± 0.016 | 0.348 ± 0.012 | 0.410 ± 0.015 | 0.320 ± 0.018 |
| valsartan_smarts | 0.179 ± 0.358 | 0.000 ± 0.000 | 0.000 ± 0.000 | 0.000 ± 0.000 | 0.000 ± 0.000 |
| zaleplon_mpo | 0.358 ± 0.062 | 0.346 ± 0.032 | 0.333 ± 0.026 | 0.221 ± 0.072 | 0.325 ± 0.027 |
| Sum | 14.196 | 13.751 | 13.471 | 13.156 | 13.024 |

Table 14: **Goal-directed hit generation results** (continued).

| Oracle | LSTM HC | SMILES-GA | SynNet | DoG-Gen | DST |
|---|---|---|---|---|---|
| albuterol_similarity | 0.719 ± 0.018 | 0.661 ± 0.066 | 0.584 ± 0.039 | 0.676 ± 0.013 | 0.619 ± 0.020 |
| amlodipine_mpo | 0.593 ± 0.016 | 0.549 ± 0.009 | 0.565 ± 0.007 | 0.536 ± 0.003 | 0.516 ± 0.007 |
| celecoxib_rediscovery | 0.539 ± 0.018 | 0.344 ± 0.027 | 0.441 ± 0.027 | 0.464 ± 0.009 | 0.380 ± 0.006 |
| deco_hop | 0.826 ± 0.017 | 0.611 ± 0.006 | 0.613 ± 0.009 | 0.800 ± 0.007 | 0.608 ± 0.008 |
| drd2 | 0.919 ± 0.015 | 0.908 ± 0.019 | 0.969 ± 0.004 | 0.948 ± 0.001 | 0.820 ± 0.014 |
| fexofenadine_mpo | 0.725 ± 0.003 | 0.721 ± 0.015 | 0.761 ± 0.015 | 0.695 ± 0.003 | 0.725 ± 0.005 |
| gsk3b | 0.839 ± 0.015 | 0.629 ± 0.044 | 0.789 ± 0.032 | 0.831 ± 0.021 | 0.671 ± 0.032 |
| isomers_c7h8n2o2 | 0.485 ± 0.045 | 0.913 ± 0.021 | 0.455 ± 0.031 | 0.465 ± 0.018 | 0.548 ± 0.069 |
| isomers_c9h10n2o2pf2cl | 0.342 ± 0.027 | 0.860 ± 0.065 | 0.241 ± 0.064 | 0.199 ± 0.016 | 0.458 ± 0.063 |
| jnk3 | 0.661 ± 0.039 | 0.316 ± 0.022 | 0.630 ± 0.034 | 0.595 ± 0.023 | 0.556 ± 0.057 |
| median1 | 0.255 ± 0.010 | 0.192 ± 0.012 | 0.218 ± 0.008 | 0.217 ± 0.001 | 0.232 ± 0.009 |
| median2 | 0.248 ± 0.008 | 0.198 ± 0.005 | 0.235 ± 0.006 | 0.212 ± 0.000 | 0.185 ± 0.020 |
| mestranol_similarity | 0.526 ± 0.032 | 0.469 ± 0.029 | 0.399 ± 0.021 | 0.437 ± 0.007 | 0.450 ± 0.027 |
| osimertinib_mpo | 0.796 ± 0.002 | 0.817 ± 0.011 | 0.796 ± 0.003 | 0.774 ± 0.002 | 0.785 ± 0.004 |
| perindopril_mpo | 0.489 ± 0.007 | 0.447 ± 0.013 | 0.557 ± 0.011 | 0.474 ± 0.002 | 0.462 ± 0.008 |
| qed | 0.939 ± 0.000 | 0.940 ± 0.000 | 0.941 ± 0.000 | 0.934 ± 0.000 | 0.938 ± 0.000 |
| ranolazine_mpo | 0.714 ± 0.008 | 0.699 ± 0.026 | 0.741 ± 0.010 | 0.711 ± 0.006 | 0.632 ± 0.054 |
| scaffold_hop | 0.533 ± 0.012 | 0.494 ± 0.011 | 0.502 ± 0.012 | 0.515 ± 0.005 | 0.497 ± 0.004 |
| sitagliptin_mpo | 0.066 ± 0.019 | 0.363 ± 0.057 | 0.025 ± 0.014 | 0.048 ± 0.008 | 0.075 ± 0.032 |
| thiothixene_rediscovery | 0.438 ± 0.008 | 0.315 ± 0.017 | 0.401 ± 0.019 | 0.375 ± 0.004 | 0.366 ± 0.006 |
| troglitazone_rediscovery | 0.354 ± 0.016 | 0.263 ± 0.024 | 0.283 ± 0.008 | 0.416 ± 0.019 | 0.279 ± 0.019 |
| valsartan_smarts | 0.000 ± 0.000 | 0.000 ± 0.000 | 0.000 ± 0.000 | 0.000 ± 0.000 | 0.000 ± 0.000 |
| zaleplon_mpo | 0.206 ± 0.006 | 0.334 ± 0.041 | 0.341 ± 0.011 | 0.123 ± 0.016 | 0.176 ± 0.045 |
| Sum | 12.223 | 12.054 | 11.498 | 11.456 | 10.989 |

| Oracle | MARS | MIMOSA | MolPal | SELFIES-LSTM HC | DoG-AE |
|---|---|---|---|---|---|
| albuterol_similarity | 0.597 ± 0.124 | 0.618 ± 0.017 | 0.609 ± 0.002 | 0.664 ± 0.030 | 0.533 ± 0.034 |
| amlodipine_mpo | 0.504 ± 0.016 | 0.543 ± 0.003 | 0.582 ± 0.008 | 0.532 ± 0.004 | 0.507 ± 0.005 |
| celecoxib_rediscovery | 0.379 ± 0.060 | 0.393 ± 0.010 | 0.415 ± 0.001 | 0.385 ± 0.008 | 0.355 ± 0.012 |
| deco_hop | 0.589 ± 0.003 | 0.619 ± 0.003 | 0.643 ± 0.005 | 0.590 ± 0.001 | 0.765 ± 0.055 |
| drd2 | 0.891 ± 0.020 | 0.799 ± 0.017 | 0.783 ± 0.009 | 0.729 ± 0.034 | 0.943 ± 0.009 |
| fexofenadine_mpo | 0.711 ± 0.006 | 0.706 ± 0.011 | 0.685 ± 0.000 | 0.693 ± 0.004 | 0.679 ± 0.017 |
| gsk3b | 0.552 ± 0.037 | 0.554 ± 0.042 | 0.555 ± 0.011 | 0.423 ± 0.018 | 0.601 ± 0.091 |
| isomers_c7h8n2o2 | 0.728 ± 0.027 | 0.564 ± 0.046 | 0.484 ± 0.006 | 0.587 ± 0.031 | 0.239 ± 0.077 |
| isomers_c9h10n2o2pf2cl | 0.581 ± 0.013 | 0.303 ± 0.046 | 0.164 ± 0.003 | 0.352 ± 0.019 | 0.049 ± 0.015 |
| jnk3 | 0.489 ± 0.095 | 0.360 ± 0.063 | 0.339 ± 0.009 | 0.207 ± 0.013 | 0.469 ± 0.138 |
| median1 | 0.207 ± 0.011 | 0.243 ± 0.005 | 0.249 ± 0.001 | 0.239 ± 0.009 | 0.171 ± 0.009 |
| median2 | 0.181 ± 0.011 | 0.214 ± 0.002 | 0.230 ± 0.000 | 0.205 ± 0.005 | 0.182 ± 0.006 |
| mestranol_similarity | 0.388 ± 0.026 | 0.438 ± 0.015 | 0.564 ± 0.004 | 0.446 ± 0.009 | 0.370 ± 0.014 |
| osimertinib_mpo | 0.777 ± 0.006 | 0.788 ± 0.014 | 0.779 ± 0.000 | 0.780 ± 0.005 | 0.750 ± 0.012 |
| perindopril_mpo | 0.462 ± 0.006 | 0.490 ± 0.011 | 0.467 ± 0.002 | 0.448 ± 0.006 | 0.432 ± 0.013 |
| qed | 0.930 ± 0.003 | 0.939 ± 0.000 | 0.940 ± 0.000 | 0.938 ± 0.000 | 0.926 ± 0.003 |
| ranolazine_mpo | 0.740 ± 0.010 | 0.640 ± 0.015 | 0.457 ± 0.005 | 0.614 ± 0.010 | 0.689 ± 0.015 |
| scaffold_hop | 0.469 ± 0.004 | 0.507 ± 0.015 | 0.494 ± 0.000 | 0.472 ± 0.002 | 0.489 ± 0.010 |
| sitagliptin_mpo | 0.016 ± 0.003 | 0.102 ± 0.023 | 0.043 ± 0.001 | 0.116 ± 0.012 | 0.009 ± 0.005 |
| thiothixene_rediscovery | 0.344 ± 0.022 | 0.347 ± 0.018 | 0.339 ± 0.001 | 0.339 ± 0.009 | 0.314 ± 0.015 |
| troglitazone_rediscovery | 0.256 ± 0.016 | 0.299 ± 0.009 | 0.268 ± 0.000 | 0.257 ± 0.002 | 0.259 ± 0.016 |
| valsartan_smarts | 0.000 ± 0.000 | 0.000 ± 0.000 | 0.000 ± 0.000 | 0.000 ± 0.000 | 0.000 ± 0.000 |
| zaleplon_mpo | 0.187 ± 0.046 | 0.172 ± 0.036 | 0.168 ± 0.003 | 0.218 ± 0.020 | 0.049 ± 0.027 |
| Sum | 10.989 | 10.651 | 10.268 | 10.246 | 9.790 |

Table 15: **Goal-directed hit generation results** (continued).

| Oracle | GFlowNet | GA+D | SELFIES-VAE BO | Screening | SMILES-VAE BO |
|---|---|---|---|---|---|
| albuterol_similarity | 0.447 ± 0.012 | 0.495 ± 0.025 | 0.494 ± 0.012 | 0.483 ± 0.006 | 0.489 ± 0.007 |
| amlodipine_mpo | 0.444 ± 0.004 | 0.400 ± 0.032 | 0.516 ± 0.005 | 0.535 ± 0.001 | 0.533 ± 0.009 |
| celecoxib_rediscovery | 0.327 ± 0.004 | 0.223 ± 0.025 | 0.326 ± 0.007 | 0.351 ± 0.005 | 0.354 ± 0.002 |
| deco_hop | 0.583 ± 0.002 | 0.550 ± 0.005 | 0.579 ± 0.001 | 0.590 ± 0.001 | 0.589 ± 0.001 |
| drd2 | 0.590 ± 0.070 | 0.382 ± 0.205 | 0.569 ± 0.039 | 0.545 ± 0.015 | 0.555 ± 0.043 |
| fexofenadine_mpo | 0.693 ± 0.006 | 0.587 ± 0.007 | 0.670 ± 0.004 | 0.666 ± 0.004 | 0.671 ± 0.003 |
| gsk3b | 0.651 ± 0.026 | 0.342 ± 0.019 | 0.350 ± 0.034 | 0.438 ± 0.034 | 0.386 ± 0.006 |
| isomers_c7h8n2o2 | 0.366 ± 0.043 | 0.854 ± 0.015 | 0.325 ± 0.028 | 0.168 ± 0.034 | 0.161 ± 0.017 |
| isomers_c9h10n2o2pf2cl | 0.110 ± 0.031 | 0.657 ± 0.020 | 0.200 ± 0.030 | 0.106 ± 0.021 | 0.084 ± 0.009 |
| jnk3 | 0.440 ± 0.022 | 0.219 ± 0.021 | 0.208 ± 0.022 | 0.238 ± 0.024 | 0.241 ± 0.026 |
| median1 | 0.202 ± 0.004 | 0.180 ± 0.009 | 0.201 ± 0.003 | 0.205 ± 0.005 | 0.202 ± 0.006 |
| median2 | 0.180 ± 0.000 | 0.121 ± 0.005 | 0.185 ± 0.001 | 0.200 ± 0.004 | 0.195 ± 0.001 |
| mestranol_similarity | 0.322 ± 0.007 | 0.371 ± 0.016 | 0.386 ± 0.009 | 0.409 ± 0.019 | 0.399 ± 0.005 |
| osimertinib_mpo | 0.784 ± 0.001 | 0.672 ± 0.027 | 0.765 ± 0.002 | 0.764 ± 0.001 | 0.771 ± 0.002 |
| perindopril_mpo | 0.430 ± 0.010 | 0.172 ± 0.088 | 0.429 ± 0.003 | 0.445 ± 0.004 | 0.442 ± 0.004 |
| qed | 0.921 ± 0.004 | 0.860 ± 0.014 | 0.936 ± 0.001 | 0.938 ± 0.000 | 0.938 ± 0.000 |
| ranolazine_mpo | 0.652 ± 0.002 | 0.555 ± 0.015 | 0.452 ± 0.025 | 0.411 ± 0.010 | 0.457 ± 0.012 |
| scaffold_hop | 0.463 ± 0.002 | 0.413 ± 0.009 | 0.455 ± 0.004 | 0.471 ± 0.002 | 0.470 ± 0.003 |
| sitagliptin_mpo | 0.008 ± 0.003 | 0.281 ± 0.022 | 0.084 ± 0.015 | 0.022 ± 0.003 | 0.023 ± 0.004 |
| thiothixene_rediscovery | 0.285 ± 0.012 | 0.223 ± 0.029 | 0.297 ± 0.004 | 0.317 ± 0.003 | 0.317 ± 0.007 |
| troglitazone_rediscovery | 0.188 ± 0.001 | 0.152 ± 0.013 | 0.243 ± 0.004 | 0.249 ± 0.003 | 0.257 ± 0.003 |
| valsartan_smarts | 0.000 ± 0.000 | 0.000 ± 0.000 | 0.002 ± 0.003 | 0.000 ± 0.000 | 0.002 ± 0.004 |
| zaleplon_mpo | 0.035 ± 0.030 | 0.244 ± 0.015 | 0.206 ± 0.015 | 0.072 ± 0.014 | 0.039 ± 0.012 |
| Sum | 9.131 | 8.964 | 8.887 | 8.635 | 8.587 |

| Oracle | Pasithea | GFlowNet-AL | JT-VAE BO | Graph MCTS | MolDQN |
|---|---|---|---|---|---|
| albuterol_similarity | 0.447 ± 0.007 | 0.390 ± 0.008 | 0.485 ± 0.029 | 0.580 ± 0.023 | 0.320 ± 0.015 |
| amlodipine_mpo | 0.504 ± 0.003 | 0.428 ± 0.002 | 0.519 ± 0.009 | 0.447 ± 0.008 | 0.311 ± 0.008 |
| celecoxib_rediscovery | 0.312 ± 0.007 | 0.257 ± 0.003 | 0.299 ± 0.009 | 0.264 ± 0.013 | 0.099 ± 0.005 |
| deco_hop | 0.579 ± 0.001 | 0.583 ± 0.001 | 0.585 ± 0.002 | 0.554 ± 0.002 | 0.546 ± 0.001 |
| drd2 | 0.255 ± 0.040 | 0.468 ± 0.046 | 0.506 ± 0.136 | 0.300 ± 0.050 | 0.025 ± 0.001 |
| fexofenadine_mpo | 0.660 ± 0.015 | 0.688 ± 0.002 | 0.667 ± 0.010 | 0.574 ± 0.009 | 0.478 ± 0.012 |
| gsk3b | 0.281 ± 0.038 | 0.588 ± 0.015 | 0.350 ± 0.051 | 0.281 ± 0.022 | 0.241 ± 0.008 |
| isomers_c7h8n2o2 | 0.673 ± 0.030 | 0.241 ± 0.055 | 0.103 ± 0.016 | 0.530 ± 0.035 | 0.431 ± 0.035 |
| isomers_c9h10n2o2pf2cl | 0.345 ± 0.145 | 0.064 ± 0.012 | 0.090 ± 0.035 | 0.454 ± 0.067 | 0.342 ± 0.026 |
| jnk3 | 0.154 ± 0.018 | 0.362 ± 0.021 | 0.222 ± 0.009 | 0.110 ± 0.019 | 0.111 ± 0.008 |
| median1 | 0.178 ± 0.009 | 0.190 ± 0.002 | 0.179 ± 0.003 | 0.195 ± 0.005 | 0.122 ± 0.007 |
| median2 | 0.179 ± 0.004 | 0.173 ± 0.001 | 0.180 ± 0.003 | 0.132 ± 0.002 | 0.088 ± 0.003 |
| mestranol_similarity | 0.361 ± 0.016 | 0.295 ± 0.004 | 0.356 ± 0.013 | 0.281 ± 0.008 | 0.188 ± 0.007 |
| osimertinib_mpo | 0.749 ± 0.007 | 0.787 ± 0.003 | 0.775 ± 0.004 | 0.700 ± 0.004 | 0.674 ± 0.006 |
| perindopril_mpo | 0.421 ± 0.008 | 0.421 ± 0.002 | 0.430 ± 0.009 | 0.277 ± 0.013 | 0.213 ± 0.043 |
| qed | 0.931 ± 0.002 | 0.902 ± 0.005 | 0.934 ± 0.002 | 0.892 ± 0.006 | 0.731 ± 0.018 |
| ranolazine_mpo | 0.347 ± 0.012 | 0.632 ± 0.007 | 0.508 ± 0.055 | 0.239 ± 0.027 | 0.051 ± 0.020 |
| scaffold_hop | 0.456 ± 0.003 | 0.460 ± 0.002 | 0.470 ± 0.005 | 0.412 ± 0.003 | 0.405 ± 0.004 |
| sitagliptin_mpo | 0.088 ± 0.013 | 0.006 ± 0.001 | 0.046 ± 0.027 | 0.056 ± 0.012 | 0.003 ± 0.002 |
| thiothixene_rediscovery | 0.288 ± 0.006 | 0.266 ± 0.005 | 0.282 ± 0.008 | 0.231 ± 0.004 | 0.099 ± 0.007 |
| troglitazone_rediscovery | 0.240 ± 0.002 | 0.186 ± 0.003 | 0.237 ± 0.005 | 0.224 ± 0.009 | 0.122 ± 0.004 |
| valsartan_smarts | 0.006 ± 0.012 | 0.000 ± 0.000 | 0.000 ± 0.000 | 0.000 ± 0.000 | 0.000 ± 0.000 |
| zaleplon_mpo | 0.091 ± 0.013 | 0.010 ± 0.001 | 0.125 ± 0.038 | 0.058 ± 0.019 | 0.010 ± 0.005 |
| Sum | 8.556 | 8.406 | 8.358 | 7.803 | 5.620 |

Table 16: **Ablation study on goal-directed hit generation.** The results are the means and standard deviations of PMO AUC top-10 of 3 runs. The best results are highlighted in bold.

| Oracle | Attaching (A) | A + Token remasking | A + GPT remasking | A + Frag. remasking (F) | A + F + MCG |
|---|---|---|---|---|---|
| albuterol_similarity | $0.872 \pm 0.032$ | $0.895 \pm 0.033$ | $0.908 \pm 0.039$ | $0.932 \pm 0.007$ | $\mathbf{0.937} \pm 0.010$ |
| amlodipine_mpo | $0.769 \pm 0.029$ | $0.802 \pm 0.016$ | $0.780 \pm 0.032$ | $0.804 \pm 0.006$ | $\mathbf{0.810} \pm 0.012$ |
| celecoxib_rediscovery | $\mathbf{0.859} \pm 0.008$ | $0.821 \pm 0.010$ | $0.847 \pm 0.006$ | $0.826 \pm 0.018$ | $0.826 \pm 0.018$ |
| deco_hop | $0.917 \pm 0.009$ | $0.945 \pm 0.006$ | $0.955 \pm 0.005$ | $0.953 \pm 0.016$ | $\mathbf{0.960} \pm 0.010$ |
| drd2 | $\mathbf{0.995} \pm 0.000$ | $\mathbf{0.995} \pm 0.000$ | $\mathbf{0.995} \pm 0.000$ | $\mathbf{0.995} \pm 0.000$ | $\mathbf{0.995} \pm 0.000$ |
| fexofenadine_mpo | $0.875 \pm 0.019$ | $0.886 \pm 0.017$ | $\mathbf{0.905} \pm 0.012$ | $0.894 \pm 0.028$ | $0.894 \pm 0.028$ |
| gsk3b | $0.985 \pm 0.003$ | $0.985 \pm 0.003$ | $\mathbf{0.986} \pm 0.003$ | $\mathbf{0.986} \pm 0.003$ | $\mathbf{0.986} \pm 0.001$ |
| isomers_c7h8n2o2 | $0.897 \pm 0.016$ | $0.934 \pm 0.003$ | $0.915 \pm 0.008$ | $0.934 \pm 0.002$ | $\mathbf{0.942} \pm 0.004$ |
| isomers_c9h10n2o2pf2cl | $0.816 \pm 0.025$ | $0.830 \pm 0.016$ | $0.820 \pm 0.018$ | $\mathbf{0.833} \pm 0.014$ | $\mathbf{0.833} \pm 0.014$ |
| jnk3 | $0.845 \pm 0.035$ | $0.848 \pm 0.016$ | $0.840 \pm 0.021$ | $0.856 \pm 0.016$ | $\mathbf{0.906} \pm 0.023$ |
| median1 | $0.397 \pm 0.000$ | $0.397 \pm 0.000$ | $0.396 \pm 0.000$ | $0.397 \pm 0.000$ | $\mathbf{0.398} \pm 0.000$ |
| median2 | $0.349 \pm 0.004$ | $0.350 \pm 0.006$ | $0.353 \pm 0.004$ | $0.355 \pm 0.003$ | $\mathbf{0.359} \pm 0.004$ |
| mestranol_similarity | $0.970 \pm 0.004$ | $0.980 \pm 0.002$ | $0.980 \pm 0.002$ | $0.981 \pm 0.003$ | $\mathbf{0.982} \pm 0.000$ |
| osimertinib_mpo | $\mathbf{0.876} \pm 0.003$ | $\mathbf{0.876} \pm 0.008$ | $0.873 \pm 0.003$ | $\mathbf{0.876} \pm 0.008$ | $\mathbf{0.876} \pm 0.008$ |
| perindopril_mpo | $0.697 \pm 0.014$ | $0.703 \pm 0.006$ | $0.703 \pm 0.000$ | $0.703 \pm 0.009$ | $\mathbf{0.718} \pm 0.012$ |
| qed | $0.927 \pm 0.000$ | $\mathbf{0.942} \pm 0.000$ | $0.941 \pm 0.000$ | $\mathbf{0.942} \pm 0.000$ | $\mathbf{0.942} \pm 0.000$ |
| ranolazine_mpo | $0.809 \pm 0.009$ | $0.818 \pm 0.016$ | $\mathbf{0.824} \pm 0.012$ | $0.821 \pm 0.011$ | $0.821 \pm 0.011$ |
| scaffold_hop | $0.617 \pm 0.002$ | $0.621 \pm 0.012$ | $0.626 \pm 0.005$ | $\mathbf{0.628} \pm 0.008$ | $\mathbf{0.628} \pm 0.008$ |
| sitagliptin_mpo | $0.573 \pm 0.006$ | $0.560 \pm 0.037$ | $0.566 \pm 0.010$ | $0.573 \pm 0.050$ | $\mathbf{0.584} \pm 0.034$ |
| thiothixene_rediscovery | $0.650 \pm 0.073$ | $0.686 \pm 0.121$ | $0.677 \pm 0.122$ | $0.687 \pm 0.125$ | $\mathbf{0.692} \pm 0.123$ |
| troglitazone_rediscovery | $0.801 \pm 0.062$ | $0.853 \pm 0.035$ | $0.832 \pm 0.052$ | $\mathbf{0.867} \pm 0.022$ | $\mathbf{0.867} \pm 0.022$ |
| valsartan_smarts | $0.739 \pm 0.043$ | $0.797 \pm 0.036$ | $0.764 \pm 0.038$ | $0.797 \pm 0.033$ | $\mathbf{0.822} \pm 0.042$ |
| zaleplon_mpo | $0.406 \pm 0.002$ | $0.569 \pm 0.014$ | $\mathbf{0.586} \pm 0.011$ | $0.569 \pm 0.005$ | $0.584 \pm 0.011$ |
| Sum | 17.641 | 18.091 | 18.074 | 18.208 | **18.362** |

Table 17: **Ablation study on goal-directed lead optimization (kcal/mol)** with $\delta = 0.4$. The results are the mean docking scores of the most optimized leads of 3 runs. Lower is better and the best results are highlighted in bold.

| Target protein | Seed score | Attaching (A) | A + Token remasking | A + GPT remasking | A + Frag. remasking (F) | A + F + MCG |
|---|---|---|---|---|---|---|
| parp1 | -7.3 | -8.2 | -8.7 | -10.4 | **-10.6** | **-10.6** |
| | -7.8 | -8.3 | -8.2 | **-11.5** | -11.0 | -11.0 |
| | -8.2 | - | -10.9 | -11.1 | -10.9 | **-11.3** |
| fa7 | -6.4 | -7.2 | -7.8 | -8.1 | **-8.4** | **-8.4** |
| | -6.7 | -8.3 | -8.0 | -8.1 | **-8.4** | **-8.4** |
| | -8.5 | - | - | - | - | - |
| 5ht1b | -4.5 | - | -12.8 | -12.1 | **-12.9** | **-12.9** |
| | -7.6 | -11.9 | -11.8 | -12.1 | **-12.3** | **-12.3** |
| | -9.8 | - | -11.3 | -11.2 | **-11.8** | **-11.8** |
| braf | -9.3 | -9.7 | -10.7 | **-10.8** | **-10.8** | **-10.8** |
| | -9.4 | - | **-10.8** | -10.1 | -10.2 | **-10.8** |
| | -9.8 | - | -10.5 | **-10.6** | **-10.6** | **-10.6** |
| jak2 | -7.7 | -8.6 | -8.7 | **-10.2** | -10.0 | **-10.2** |
| | -8.0 | -8.9 | -9.0 | **-10.1** | -9.9 | -10.0 |
| | -8.6 | - | -9.2 | **-9.8** | -9.5 | **-9.8** |

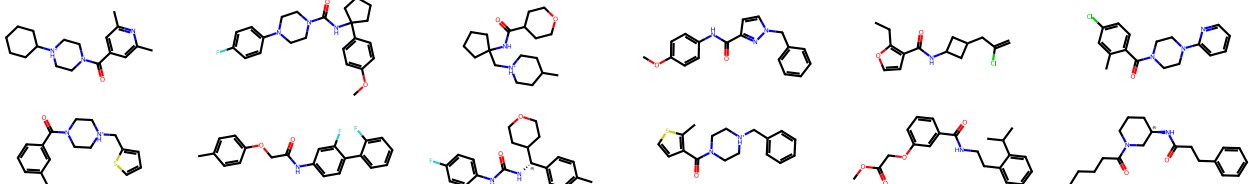

Figure 7: **Examples of generated molecules on *de novo* generation.**

| Task | Input | Generated molecules |
|---|---|---|
| Linker design & Scaffold morphing | | |
| Motif extension | | |
| Scaffold decoration | | |
| Superstructure generation | | |

Figure 8: **Examples of generated molecules on fragment-constrained generation** of Eliglustat.

