# OpenReview forum: "GenMol: A Drug Discovery Generalist with Discrete Diffusion"
_ICML.cc/2025/Conference — ICML 2025 poster_

### Official Review · Reviewer_9oFx · 2025-03-06

**Overall Recommendation:** 3

**Summary:**

The paper focuses on drug discovery and introduces a Generalist Molecular Generative Model (GenMol), a new discrete diffusion model for molecular generation. This approach has the potential to be applied to various drug-related tasks, including de novo generation, linker generation, and hit/lead generation. The model represents molecules as sequential data using fragment-based SMILES strings, known as SAFE strings. According to Figure 2, the work still utilizes atom-level tokens, which may be somewhat counterintuitive, as one might expect fragments to be used as tokens. The discrete diffusion model operates through a masking (forward) and unmasking (reverse) process.

For goal-directed generation, the work incorporates fragment remasking and molecular context guidance to enhance generation quality. Experiments are conducted on (1) De novo generation, (2) fragment-based generation, (3) hit generation using the Practical Molecular Optimization benchmark, and (4) lead generation for five protein targets. The results, along with ablation studies, suggest that the proposed approach is promising. However, further clarification on certain aspects of the methodology and evaluation could strengthen the manuscript.

**Claims And Evidence:**

1. It is unclear how many diffusion timesteps are used, making the claims about efficiency unconvincing. These claims appear around Lines 92–94, 200–208, and 634. For example, the diffusion timestep could be set to 500 or 1000, which is a common choice for graph and image diffusion models. But an autoregressive model would perform decoding in a number of steps equal to the maximum number of nodes, which is typically smaller than 500. Additionally, one advantage of autoregressive models is that they do not necessarily need to consider the entire molecular context during generation, whereas GenMol does. This raises further concerns regarding efficiency. The authors should explicitly state the hyperparameters used and provide a comparison of generation times.

2. The claim around Lines 93–96 that discrete diffusion models enable GenMol to explore chemical space with remasking should also apply to autoregressive models. While the claim holds for GenMol, it is somewhat unclear why this point is specifically listed as an advantage (Line 88). Further clarification would be helpful.

3. Regarding Lines 99–102, the authors suggest that using fragments as units aligns better with chemists’ intuition for chemical space exploration. If this is the case, why are fragments not used as tokens in the discrete model? It also introduces an additional concern in the following point.

4. The manuscript lacks a convincing discussion on why SAFE is chosen over SMILES, given that the discrete diffusion model operates at the atom level. In other words, when extending discrete diffusion models to molecular sequential strings, one might first consider SMILES as a more direct option. Is there a specific reason for opting for SAFE instead of SMILES? While the manuscript discusses fragment-based remasking, this concept could also be applied to SMILES, as SAFE is derived from SMILES, and the molecular structures represented by SAFE could also be represented using SMILES.

**Essential References Not Discussed:**

There is a lack of survey and discussion on discrete molecular diffusion models. There  are many graph-based discrete diffusion models, the authors should discuss their efficiency and performance and compare them in experiments if necessary. For example, the authors have discussed about DiGress, which could be used for fragment-based generation as presented in the original paper, but it is not tested in Table 2.

**Experimental Designs Or Analyses:**

1. The authors should provide data statistics for the dataset used to pretrain GenMol.

2. In molecular and fragment-based generation, only one baseline is included. To ensure a comprehensive evaluation, the authors should consider testing additional baselines, including discrete diffusion models, graph diffusion models, and autoregressive models (e.g., SMILES-based models). This would help demonstrate the advantages of SAFE-based discrete diffusion in a fair comparison.

3. The benefits of the method proposed in Section 4 (or Figure 3a) could also be applicable to SMILES-based generation. Could the approach be adapted to other baselines? Discussing this possibility would enhance the generalizability of the proposed method.

4. There are several missing values in Table 4.

**Methods And Evaluation Criteria:**

1. The details regarding tokenization are unclear. The vocabulary size is stated as 1880 (Line 726), but it is not explained how this vocabulary is constructed. Additionally, it is not discussed whether this vocabulary can generalize to broader tasks, which is important given the claim about the model being a generalist. Clarification on these aspects would be beneficial.

2. The fragment vocabulary derived from the dataset is not sufficiently discussed. The BRICS algorithm is briefly mentioned in Line 177, and I assume it serves as the underlying algorithm for defining fragments. However, BRICS tends to prioritize synthetically relevant fragments. As noted in previous work [1], BRICS has several limitations: it breaks bonds based on a limited set of chemical reaction rules, tends to generate a few large fragments, creates variations of the same underlying structures, and results in a very large vocabulary with many infrequent fragments.

3. It is unclear how the iterations in Figure 3 contribute to the quality and efficiency of generation. Additional analysis on this aspect would be helpful to better understand its impact.

4. Figure 3 shows that the framework heavily relies on the scoring oracle. What happens if the oracle is not available? Additionally, if the function used for the oracle does not provide accurate scores, how would this affect performance? Does the proposed GenMol model still perform well under these conditions?

Reference:

[1] Motif-based Graph Self-Supervised Learning for Molecular Property Prediction. NeurIPS 2021.

**Other Comments Or Suggestions:**

NA

**Other Strengths And Weaknesses:**

The contributions of this work are multifaceted, including the GenMol model, which refers to the checkpoint used in the experiments and is expected to function as a generalist model. Additionally, the work introduces the GenMol framework (e.g., Figure 3a), which appears to have the potential for generalization to other models. However, the authors use the same name for both the model and the framework, which at times makes the manuscript difficult to follow. It is unclear whether certain benefits discussed apply specifically to the model or to the framework.

For instance, the authors present discrete diffusion as a major contribution (Lines 80–108), but it does not appear to introduce any new developments to the discrete diffusion model itself (as described in Section 4). Using discrete diffusion models for molecules is not a new idea as well.

Furthermore, does Section 5.1 focus on unconditional generation? If so, how can the proposed GenMol framework in Figure 3a be adapted to this problem?

The goal of this work is to develop a generalist model, and accordingly, it has been tested on multiple tasks with results presented in various tables. However, the selection of baselines for each task may not be comprehensive. For example, in de novo generation and fragment-based generation, only one baseline is included, and for lead optimization, only two baselines are considered. Expanding the comparison to include more relevant baselines would strengthen the evaluation.

**Questions For Authors:**

1. How is the molecular context defined in Lines 106–107?

2. Could the fragment remasking approach be adapted to other autoregressive models?

3. The molecular optimization performance heavily depends on frequent Oracle calls (e.g., 10,000 times). What would happen if the number of calls were significantly reduced to fewer than 10 or if inaccurate models were used as Oracles?

4. Do Sections 5.1 and 5.2 utilize the framework presented in Figure 3a?

**Relation To Broader Scientific Literature:**

The contribution could aid drug discovery in the fields of chemistry and biology.

**Theoretical Claims:**

The work presents the derivation of Eq. (7) in Appendix C. I have no issues about it.

---

> ### Author Rebuttal · Authors · 2025-04-01
>
> We sincerely appreciate your comments that our contribution could aid drug discovery and GenMol is a generalist. We address your concerns below.
>
> ---
> > One baseline in de novo and fragment-based generation.
>
> We included JT-VAE (Jin et al., 2018) in Table 9, and have provided the results of DiGress in our *first response to 5s8Z*.
>
> ---
> > No new developments in discrete diffusion. Graph discrete diffusion should be discussed.
>
> **GenMol is the first discrete diffusion framework on small molecule sequences**, which is superior to graph discrete diffusion that suffers from low validity and efficiency (GenMol vs. DiGress above). We also showed **confidence sampling achieves better quality and efficiency in molecular sequence generation**. Moreover, we proposed **MCG tailored for discrete sequence diffusion**.
>
> ---
> > It is unclear how many diffusion timesteps are used.
>
> GenMol uses confidence sampling with $N$ denoting the number of tokens to unmask at each step (line 323). GenMol with $N=1$ uses the same number of steps with AR models, and higher $N$ improves efficiency. In Table 1, GenMol with $N=3$ shows higher quality than the AR model with 2.5x faster sampling.
>
> ---
> > Could the fragment remasking be adapted to AR models?
>
> Yes, but it cannot be applied without relying on a specific generation order in AR models, leading to inferior performance (Table 5).
>
> ---
> > Why are fragments not used as tokens?
>
> Using fragments as tokens is challenging due to **the large vocabulary, limited generalizability, and the difficulty of tokenizing attachment points**. Applying BRICS to ZINC250k yields 39034 fragments, resulting in an imbalance with sequence lengths of 3~5. It also does not allow the model to generate novel fragments. We have introduced **a simpler yet effective approach that generates tokens at atom level and performs the remasking at fragment level**.
>
> ---
> > A reason for using SAFE instead of SMILES? Could the approach be adapted to SMILES generation?
>
> **SAFE makes fragment remasking simpler and more effective**, and adapting the approach to SMILES is non-trivial. As tokens that belong to the same fragment are not contiguous in SMILES, SMILES models need to classify each token into fragments before every remasking and cannot guarantee the re-predicted tokens belong to the same fragment. On the other hand, SAFE allows GenMol to easily identify fragments in a sequence and generate new fragments.
>
> ---
> > Tokenization details are unclear.
>
> We adopt the SAFE tokenizer (Noutahi et al., 2024) (lines 725-726), which is a BPE tokenizer with the SMILES regular expression. It covers 1873 different atoms, bonds, and ions, and thus can be universally used for molecule-related tasks.
>
> ---
> > Do Sections 5.1 and 5.2 utilize the framework in Figure 3a? It is unclear how the iterations in Figure 3 contribute to the quality and efficiency of generation.
>
> No, Sections 5.1 and 5.2 use the process of Figure 2(c1-c5) (line 173, Figure 3(a) caption). The generic quality and efficiency stem from the discrete diffusion process, and the iterations in Figure 3(a) are multiple rounds of the process, not the process itself.
>
> ---
> > What if the number of calls is low or the oracle is incorrect?
>
> We followed Gao et al., 2022 in Section 5.3, where the 10k oracle calls were set for all baselines. Nevertheless, we observed that **GenMol consistently performed better in both low and high oracle call regimes**, so we hypothesize that GenMol would perform better with fewer calls or an incorrect oracle, and will include this in the revision.
>
> ---
> > Only two baselines in lead optimization. Missing values in Table 4.
>
> In Table 4, baseline selection and the use of the ‘-’ symbol to indicate failure (no successful leads generated) followed Wang et al., 2023 (lines 412-413).
>
> ---
> > The same name was used for the model and the framework.
>
> **GenMol is the framework that uses discrete diffusion in versatile ways for diverse drug discovery tasks**. GenMol adopts the sampling of Figure 2(c1-c5) for *de novo* and fragment-constrained generation and adds fragment remasking of Figure 3(a) for goal-directed optimization.
>
> ---
> > How is the molecular context defined?
>
> See our *second response to 5s8Z*.
>
> ---
> > The fragment vocabulary is not sufficiently discussed.
>
> The training SAFEs are based on BRICS following Noutahi et al., 2024. During inference, we used two other rules (lines 234-236, 756-762): $R_{vocab}$, that cuts one non-ring single bond, to construct the vocabulary, and $R_{remask}$, that cuts all non-ring single bonds, to determine which fragments the remasking will operate on.
>
> ---
> > Data statistics should be provided.
>
> We used the dataset of Noutahi et al., 2024, and will include the statistics in the revision.
>
> ---
> We hope our response could address your questions and clarify any confusion. If our response is satisfactory, we would like to kindly ask you to consider raising your score. Otherwise, we will be happy to further discuss and update the paper.

---

### Official Review · Reviewer_5s8Z · 2025-03-11

**Overall Recommendation:** 5

**Summary:**

GenMol is a generalist molecular generative model that uses a masked discrete diffusion framework with a BERT-based architecture to generate SAFE molecular sequences, enabling efficient, non-autoregressive decoding and better sampling efficiency. It introduces fragment remasking to optimize molecules by selectively regenerating fragments, enhancing chemical space exploration, and employs molecular context guidance (MCG) to refine predictions using full molecular context. GenMol outperforms existing models across various drug discovery tasks, demonstrating its performance in molecular design.

## update after rebuttal

I believe the authors have adequately addressed all of my concerns. I have also reviewed the points raised by the other reviewers and continue to find this work solid, novel, and impactful. There are no remaining concerns—minor or major—from my side, so I have adjusted my score accordingly.

**Claims And Evidence:**

All of the claims are supported by extensive empirical evaluations and experiments. The paper is overall convincing and presents extensive empirical results.

**Essential References Not Discussed:**

Maybe the work can cite (Nie, S., Zhu, F., You, Z., Zhang, X., Ou, J., Hu, J., ... & Li, C. (2025). Large Language Diffusion Models. arXiv preprint arXiv:2502.09992.) as the most recent work on sequence generation with diffusion models which has amazing results and has caught attention of the community recently.

**Experimental Designs Or Analyses:**

The experimental design in the paper is generally well-structured, covering a range of drug discovery tasks to validate GenMol’s capabilities. The use of a single checkpoint without task-specific finetuning strengthens the claim of model generalization. However, I would like to mention this aspect:
**Fragment Remasking & MCG Ablations**: The ablation studies show that fragment remasking and molecular context guidance (MCG) improve performance, but it is unclear how these compare to alternative molecular generation strategies beyond SAFE-GPT. Additional baselines, such as reinforcement learning-based optimization, could provide a more comprehensive validation.
Overall, the experimental design is sound, but a more rigorous comparison with additional baselines would further strengthen the findings.

**Methods And Evaluation Criteria:**

Yes, it follows previous works' evaluation metrics and datasets to evaluate itself.

**Other Comments Or Suggestions:**

Nothing in mind.

**Other Strengths And Weaknesses:**

One of the key strengths of this paper is its **fragment-based generation approach**, which aligns well with industry needs. In practical drug discovery, designing and synthesizing molecules from predefined building blocks is significantly more efficient than synthesizing entirely new structures from scratch. This makes GenMol's fragment remasking strategy particularly valuable, as it reflects real-world workflows and can accelerate the molecular design process.

Additionally, the paper demonstrates strong **methodological innovation** by combining **masked discrete diffusion with bidirectional sequence modeling**, enabling non-autoregressive parallel decoding. This enhances both sampling efficiency and the quality of generated molecules, addressing limitations in prior autoregressive and graph-based models. The **unified generative framework** that generalizes across multiple drug discovery tasks without requiring task-specific fine-tuning further adds to its significance, as it simplifies deployment in practical settings.

On the other hand, while the paper presents extensive empirical validation, its clarity could be improved in certain sections. Some methodological details, such as how molecular context guidance (MCG) is integrated into the diffusion process, could be more explicitly explained to enhance reproducibility. Furthermore, while GenMol shows superior performance over existing methods, an **ablation study on model scalability and robustness**—such as how it performs on structurally diverse datasets beyond ZINC and UniChem (e.g. Enamine)—would further strengthen the work.

Overall, this paper makes an **original and significant contribution** by developing a **computationally efficient, chemically intuitive, and versatile generative model** for molecular design. Its alignment with both **theoretical advancements in generative modeling and practical drug discovery needs** makes it a valuable addition to the field.

**Questions For Authors:**

- What would be the cost of training this model on a broader space like Enamine 1B dataset, or even bigger datasets to fully explore drug space by leveraging this models' strengths?
- Is there any possibility of conditioning this model on a specific gene expression or the protein embedding of a specific target to guide the generation towards a specific biological target or not?

**Relation To Broader Scientific Literature:**

The key contributions of GenMol build upon and extend several existing ideas in molecular generative modeling, discrete diffusion, and drug discovery optimization. Here’s how they relate to the broader scientific literature:

1. **Discrete Diffusion for Molecular Generation**: GenMol leverages a **masked discrete diffusion model**, which aligns with prior work on discrete diffusion in natural language processing (Austin et al., 2021) and molecular modeling (Sahoo et al., 2024; Shi et al., 2024). This approach contrasts with continuous diffusion models (e.g., Hoogeboom et al., 2022), which have been more commonly used in generative chemistry. By adopting a **BERT-like bidirectional attention** mechanism, GenMol improves upon **autoregressive generative models** (e.g., Gomez-Bombarelli et al., 2018) by enabling parallel token prediction and increased efficiency.

2. **Fragment-Based Molecular Design**: The **fragment remasking strategy** extends previous work in fragment-based drug design, which has long been a key principle in medicinal chemistry (Congreve et al., 2003). While prior generative models often treated molecules as atomic graphs (Jin et al., 2018; You et al., 2018), GenMol aligns more closely with chemist intuition by using fragments as the primary unit of generation. This also improves **chemical space exploration**, addressing limitations in previous sequence-based molecular design models like SAFE-GPT (Noutahi et al., 2024).

3. **Molecular Context Guidance (MCG)**: The proposed MCG technique enhances discrete diffusion by incorporating context-dependent constraints, similar in spirit to **controlled generation in NLP** (Keskar et al., 2019) and **conditional molecular design** (Zhavoronkov et al., 2019). Unlike previous methods that relied on reinforcement learning (Olivecrona et al., 2017) or constrained optimization (Gao et al., 2022), GenMol integrates context-awareness **directly into the generative process**, enabling more targeted molecular design.

4. **Efficiency and Generalization Across Drug Discovery Tasks**: GenMol’s ability to perform **de novo generation, fragment-constrained generation, and goal-directed optimization** in a unified manner improves upon **task-specific models** such as GraphAF (Shi et al., 2020) and f-RAG (Lee et al., 2024a). Unlike previous approaches that required separate fine-tuning for different tasks, GenMol’s single checkpoint generalizes across multiple drug discovery scenarios, making it more adaptable to real-world applications.

In summary, GenMol synthesizes advances in **discrete diffusion, bidirectional sequence modeling, fragment-based design, and guided molecular generation** to offer a more **efficient, generalizable, and chemically intuitive** approach to molecular generation, surpassing limitations of existing autoregressive and fine-tuned models in drug discovery.

**Theoretical Claims:**

There is just a simple derivation of MCG which is simple and straight-forward. Thus, I have checked all of the proofs for theoretical claims.

---

> ### Author Rebuttal · Authors · 2025-04-01
>
> We sincerely appreciate your comments that all our claims are supported by extensive evaluations, that our proposed fragment-based generation approach aligns well with industry needs, and that our paper demonstrates strong methodological innovation. We address your questions below.
>
> ---
> > Additional baselines, such as reinforcement learning-based optimization, could provide a more comprehensive validation. (Also related to Reviewer 9oFx’s comment: *In de novo generation and fragment-based generation, only one baseline is included.*)
>
> We used the PMO benchmark (Gao et al., 2022) for the goal-directed hit generation task (Tables 3, 11-13), and **compared our GenMol with a total of 28 baselines**. These baselines span a broad class of molecular optimization methods, including reinforcement learning methods, genetic algorithm methods, and Bayesian optimization methods. The proposed GenMol achieves the best performance against all of these baselines and its ablated baselines (Table 5).
>
> Furthermore, here we compare GenMol with DiGress (Vignac et al., 2023), a graph discrete diffusion method on the *de novo* and fragment-constrained generation tasks, where **GenMol significantly outperforms DiGress in both quality and sampling efficiency**.
>
> **Table: *De novo* generation results.**
> | Method | Validity (%) | Uniqueness (%) | Quality (%) | Diversity (%) | Sampling time (s) |
> | --- | --- | --- | --- | --- | --- |
> | DiGress | 89.6 | 100.0 | 36.8 | 0.885 | 1241.9 |
> | GenMol ($N=1,\tau=0.5,r=0.5$) | 100.0 | 99.7 | 84.6 | 0.818 | 21.1 |
> | GenMol ($N=1,\tau=1.5,r=10$) | 95.6 | 98.3 | 39.7 | 0.911 | 20.9 |
>
> **Table: Fragment-constrained generation results.**
> | Method | Task | Validity (%) | Uniqueness (%) | Quality (%) | Diversity (%) | Distance |
> | --- | --- | --- | --- | --- | --- | --- |
> | DiGress | Linker design (scaffold morphing) | 31.2 | 84.3 | 6.1 | 0.745 | 0.724 |
> | | Motif extension | 21.8 | 94.5 | 4.2 | 0.818 | 0.794 |
> | | Scaffold decoration | 29.3 | 91.0 | 9.1 | 0.793 | 0.785 |
> | | Superstructure generation | 26.7 | 85.9 | 7.4 | 0.789 | 0.776 |
> | GenMol | Linker design (scaffold morphing) | 100.0 | 83.7 | 21.9 | 0.547 | 0.563 |
> | | Motif extension | 82.9 | 77.5 | 30.1 | 0.617 | 0.682 |
> | | Scaffold decoration | 96.6 | 82.7 | 31.8 | 0.591 | 0.651 |
> | | Superstructure generation | 97.5 | 83.6 | 34.8 | 0.599 | 0.762 |
>
> ---
> > How molecular context guidance (MCG) is integrated into the diffusion process could be more explicitly explained. (Also related to Reviewer 9oFx’s comment: *How is the molecular context defined in Lines 106-107?*)
>
> MCG is Autoguidance (Karras et al., 2024) specifically designed for the discrete diffusion of GenMol to fully utilize the given molecular context information. This is done by comparing two outputs from a single model, with good (i.e., less masked) and poor (i.e., more masked) input, respectively. Specifically, the good input $z_t$ is a given partially masked sequence, which is further masked by $100 \cdot \gamma$ % to yield poor input $\tilde{z}_t$, and the two resulting logits are compared and calibrated as Eq. (7). Intuitively, the logits with the poor input exaggerate errors in the logits with the intact input, and $w>1$ removes the errors from the good logits.
>
> Following the use cases of *context* in LLM literature, *molecular context* in MCG denotes the given information in the form of the input sequence, i.e., given fragments in fragment-constrained generation and partially masked sequences during the fragment remasking in goal-directed generation.
>
> We apologize for the brief description due to space limitations and will clarify this in a future revision.
>
> ---
> > What would be the cost of training this model on a broader space like the Enamine 1B dataset to fully explore drug space by leveraging this model's strengths?
>
> The cost of training GenMol on a new SMILES-formatted dataset comes from two sources: (1) preprocessing SMILES -> SAFE, and (2) training with new SAFE strings. SMILES to SAFE conversion of 249,455 ZINC250k molecules takes ~6 minutes. For GenMol to go through 1B training molecules with the same setting in the paper would take ~60 hours.
>
> ---
> > Is there any possibility of conditioning this model to guide the generation towards a specific biological target?
>
> **GenMol can be easily extended to be conditioned on specific biochemical conditions** by adopting any off-the-shelf conditioning methods. For example, protein pocket embeddings obtained from a pretrained protein model can be incorporated into GenMol via cross-attention to generate corresponding ligands [A, B].
>
> ---
> *References*
>
> [A] Fu et al., Fragment and geometry aware tokenization of molecules for structure-based drug design using language models, ICLR, 2025.
>
> [B] Wang et al., Token-Mol 1.0: tokenized drug design with large language model, arXiv, 2024.

---

> > ### Comment · Reviewer_5s8Z · 2025-04-02
> >
> > I thank authors for their efforts on preparing the response. I have adjusted my score accordingly.
> > All the best,

---

### Official Review · Reviewer_TVyV · 2025-03-13

**Overall Recommendation:** 3

**Summary:**

The paper introduces a versatile molecular generative model based on discrete diffusion applied to the Sequential Attachment-based Fragment Embedding (SAFE) representations. The method can address various drug discovery tasks uniformly (de novo generation, fragment-constrained generation, hit generation, and lead optimization) using a single backbone model. Following are the key contributions of the paper.

1. Present a discrete diffusion model that works for sequence generation for diverse drug discovery scenarios.
2. Introduce fragment remasking.
3. Introduces a new guidance method called Molecular Context Guidance (MCG) for fragment-constrained generation.
4. An ablation study that highlights the quantitative impact of each of the novel contributions stated above.

**Claims And Evidence:**

All the claims mentioned in the summary above are supported well by the experiments.

**Essential References Not Discussed:**

No

**Experimental Designs Or Analyses:**

The experiments are carefully designed to support the main claims of the paper.

**Methods And Evaluation Criteria:**

Yes

**Other Comments Or Suggestions:**

1. $x_{\theta}^l(z_t, t)$ is already a probability distribution over the token vocabulary, and you could introduce the temperature in the softmax used in $x_{\theta}^l(z_t, t)$ itself. Is there a specific reason for applying the softmax(log(x)) transformation again? If there is, please state it.

**Other Strengths And Weaknesses:**

The paper can be improved by comparing the proposed guidance method to existing methods in the literature like Nisonoff et al. (2024).

**Questions For Authors:**

1. Why does the diversity increase with N (for example in table 1)? I would expect the diversity to be low for higher N. Is the diversity increasing solely at the expense of validity?

2. I'm struggling to understand how MCG is implemented for goal-directed generation. All we can see from eq 7 is that the scores of the tokens are constructed with the scores when some more tokens are masked (governed by $\gamma$). Reframing the question, how is the goal value $y$ involved in eq 12 in Appendix C?

**Relation To Broader Scientific Literature:**

The paper introduces the non-autoregressive counterpart to SAFE-GPT (Noutahi et al., 2024). Both GenMol and SAFE-GPT use SAFE representations to solve various tasks in the drug discovery pipeline using a single model. The paper uses the MDLM (Sahoo et al (2024)) and introduces a new guidance method, which is inspired from autoguidance (Karras et al. (2024)).

**Theoretical Claims:**

There are no proofs as such. I checked the derivation in Appendix C.

---

> ### Author Rebuttal · Authors · 2025-04-01
>
> We sincerely thank you for your comments. We appreciate your positive comments that our paper introduces a versatile molecular generative model that can address various drug discovery tasks using a single backbone model, that all the claims are supported well by the experiments, and that the experiments are carefully designed. We address your concerns and questions below.
>
> ---
> > The paper can be improved by comparing the proposed guidance method to existing methods in the literature like Nisonoff et al. (2024). In MCG (Eq. (7)), how is the goal value $y$ involved?
>
> We appreciate the suggestion. MCG and Discrete Guidance (Nisonoff et al., 2024) are not directly comparable: MCG does not rely on the target property conditions for guidance while Discrete Guidance does. To use Discrete Guidance, one has to train a conditional generative model that is conditioned on the target properties, which cannot be easily transferred to other tasks. Instead, **MCG is specifically designed for the discrete diffusion of GenMol that is property-unconditional and general-purposed**. For example, MCG can be used for fragment-constrained generation as well as goal-directed generation, while Discrete Guidance cannot. Therefore, the two methods have different objectives and can be used orthogonally.
>
> The score $y$ (Eq. (5)) is used for fragment vocabulary selection which allows GenMol to effectively perform goal-directed generation without any target property-based diffusion guidance, and is not related to MCG.
>
> ---
> > $x^l_\theta(z_t, t)$ is already a probability distribution over the token vocabulary, and you could introduce the temperature in the softmax used in itself. Is there a specific reason for applying the softmax(log(x)) transformation again?
>
> To avoid introducing additional notation, we used $\log(x^l_\theta(z_t, t))$ to denote the logits generated by the model, and in the actual implementation, we applied the temperature $\tau$ to the generated logits during inference (Eq. (4)). We will clarify this in the revised version.
>
> ---
> > Why does the diversity increase with $N$ (in Table 1)? Is the diversity increasing solely at the expense of validity?
>
> A larger $N$ means the model predicts more tokens at once at each time step, which means the model is given fewer tokens to complete the sequence on average. The less conditional information results in more freedom in the generation, leading to an increase in diversity. On the other hand, it also puts more burden on the model capacity, which may lead to a drop in molecular validity (Table 1 and Figure 4).

---

### Decision · Program_Chairs · 2025-05-01

**Decision:**

Accept (poster)

**Comment:**

The paper proposes GenMol, a generalist molecular generative model based on a masked discrete diffusion framework with a BERT-based architecture to generate SAFE molecular sequences. It introduces fragment remasking to optimize molecules by selectively regenerating fragments, enhancing chemical space exploration, and employs molecular context guidance (MCG) to refine predictions using full molecular context. The method can address various drug discovery tasks uniformly (de novo generation, fragment-constrained generation, hit generation, and lead optimization) using a single backbone model.


Strengths:
- The paper develops a discrete diffusion model for sequence generation for diverse drug discovery scenarios.
- The paper introduces a fragment remasking technique and Molecular Context Guidance (MCG) for fragment-constrained generation
- Experiments show superior performance on multiple drug design tasks for five protein targets.

Weaknesses:
- Ablation studies on model scalability and robustness are missing.
- In molecular and fragment-based generation, only one baseline is included.
- The paper lacks discussion on SAFE vs. SMILES representations.